# Unveiling a large fraction of hidden organosulfates in ambient organic aerosol

Jialiang Ma [1] ✉, Natalie Reininger [1,2], Cunliang Zhao[3], Damian Döbler[4], Julian Rüdiger[4], Yanting Qiu [5], Florian Ungeheuer [1], Mario Simon[1], Luca D'Angelo [1], Anna Breuninger [1], Julia David [1], Yanxin Bai[6], Yushan Li[6], Ying Xue[6], Lili Li[6], Yuchen Wang[6], Stefanie Hildmann [7], Thorsten Hoffmann[7], Bangjun Liu[3], Hongya Niu[3], Zhijun Wu[5] & Alexander L. Vogel [1] ✉

Organosulfates are key compounds driving the anthropogenic enhancement of ambient organic aerosol, however, total organosulfate quantification remains elusive due to their molecular diversity and the scarcity of authentic standards. Here, we present a solid-phase extraction method that isolates organosulfates from ambient aerosol samples and enables their identification and quantification using mass spectrometry and a charged aerosol detector, respectively. We investigate ambient aerosol samples from urban China and rural Germany and quantify ~130 and ~65 chromatographically resolved organosulfates, respectively, contributing less than ~2% to the total organic matter. We find a significantly larger organosulfate fraction appearing as a broad peak in the chromatograms from the charged aerosol detector. Confirming its origin from chromatographically non-resolved organosulfates, an all-ion fragmentation experiment reveals specific sulfate-related ions. Integrating this peak, we find the contribution of organosulfates to organic aerosol is 12-17% and ~21% in samples from urban China and rural Germany, respectively. These findings emphasise the potential of sulfur emission reduction for mitigating both sulfate-related and organic aerosol pollution.

Atmospheric fine particulate matter (PM) has major implications for human health, Earth's radiative balance and air quality[1–3]. Organic aerosol (OA) is a major component in ambient PM, typically contributing 30-50% of the fine aerosol mass in the lower troposphere, though this can reach up to 90% in pristine environments[4–7]. Therefore, understanding the chemical composition and atmospheric transformation of OA is vital for identifying their sources and developing effective mitigation strategies. Due to a large variety of natural and anthropogenic emissions, transformed by intricate multiphase chemical processes, the chemical characterization of secondary OA (SOA) is extremely challenging. Organosulfates (OSs) are an important subclass of OA[8–13], comprising up to 30% of the OA[12]. They can form through different pathways such as heterogeneous reactions between OA and $SO_2$[14–18], sulfuric acid-driven condensation reactions within aerosol particles[16,19], or even through heterogeneous reactions on building surfaces[20]. The reactive uptake of semi-volatile organic compounds, particularly isoprene epoxydiols (IEPOX), onto acidic particles can result in the formation of low-volatile OSs and, consequently, add to the OA mass[9,16,21–23]. If the involved organics are of biogenic origin, $SO_2$ emissions can effectively lead to an anthropogenic enhancement

[1]Institute for Atmospheric and Environmental Sciences, Goethe University Frankfurt, Frankfurt am Main, Germany. [2]Institute of Ecology, Evolution and Diversity, Goethe University Frankfurt, Frankfurt am Main, Germany. [3]School of Earth Science and Engineering, Hebei University of Engineering, Handan, China. [4]Air Monitoring Network, German Environment Agency, Langen, Germany. [5]State Key Joint Laboratory of Environmental Simulation and Pollution Control, Peking University, Beijing, China. [6]College of Environmental Science and Engineering, Hunan University, Changsha, China. [7]Institute of Inorganic and Analytical Chemistry, Johannes Gutenberg-University, Mainz, Germany. ✉e-mail: ma@iau.uni-frankfurt.de; vogel@iau.uni-frankfurt.de

of biogenic SOA[12,24,25]. Further studies on OSs have demonstrated that they can change the hygroscopic growth of OA and even have the potential to promote heterogeneous ice nucleation[26,27].

Several studies have analyzed OSs in different environments, e.g., in the urban OA of Chinese megacities, in which rapid formation of OSs during pollution events has been observed[28–31]. However, Brüggemann et al. reported significant sampling artifacts in the detection and quantification of monoterpene-derived OSs under atmospheric conditions[32]. LeBreton et al. used a Filter Inlet for Gases and Aerosols coupled to a Chemical Ionization Mass Spectrometer to measure OSs in near-real time, and therefore minimizing sampling artifacts, at a semi-rural site in Beijing. They quantified 17 single OSs, estimating their overall contribution to OA at 2%, suggesting that OSs are not merely artifacts of the sampling process[33]. Similarly, a study conducted in Beijing using high-performance liquid chromatography (HPLC) combined with high-resolution mass spectrometry (HRMS) reported a ~4% contribution of OSs to OA[28].

In urban and rural OA, studies have shown significant seasonality in both biogenic and anthropogenic OSs at various sites, with biogenic OSs peaking in summer and anthropogenic OSs in winter[34,35]. In marine OA, in the Yellow and Bohai Seas, the contribution of biogenic OSs ranges from 0.04 to 6.9%[36]. In the central Amazon, Glasius et al. found ~0.1 and ~0.6 µg/m³ of isoprene-derived OSs downwind of Manaus during the wet and dry seasons, respectively[37]. OSs can be transported in the atmosphere over long distances, since relatively elevated OS concentrations (~20–40 ng/m³) during Arctic haze events at Svalbard have been reported, probably originating from northern Eurasia[38]. Even in Alpine ice core samples from the Fiescherhorn glacier, OSs from both biogenic and anthropogenic organics were detected in a sample that dates back to the year 1984[39]. Although the direct analysis of molecularly resolved OSs enables the determination of their contribution to total OA, a degree of uncertainty remains in using this approach since it relies on the individual detection of single molecules and the application of surrogate standards for quantification. If no authentic standard is available, Brüggemann et al. suggested camphor-10-sulfonic acid ($C_{10}H_{16}O_4S$) as an appropriate surrogate standard[40]. Nevertheless, direct quantification can easily introduce a large uncertainty due to different ionization efficiencies or by missing low-abundant or chromatographically unresolved OSs that can result in the underestimation of the total content of OSs in ambient aerosol.

Besides the direct quantification of individual OSs, indirect methods are available that quantify the total content of OSs by determining the difference between total sulfur (e.g., determined by X-ray fluorescence) and inorganic sulfate (e.g., determined by ion chromatography). These indirect approaches have suggested that OSs contribute 6–14% and up to 20% of the total sulfate at a rural Hungarian site[41] and in urban Shenzhen, China[42], respectively, and 5–10% to the organic matter (OM) in the United States[43]. Aerosol mass spectrometry (AMS) has been used to quantify OSs in real time[44–48]. Farmer et al. reported ~12% of total sulfate being present as OSs in Riverside, California, by comparing the AMS measurements of total sulfate and the ion chromatography measurements of inorganic sulfate, while the same approach at a rural station in Germany indicated that OSs can account for up to 46% of OA during single pollution episodes[47,48]. Organosulfur abundance typically follows a seasonal cycle with the formation of isoprene-derived OSs during summer, providing direct evidence for the anthropogenic enhancement of biogenic SOA, although the majority of organosulfur species remain unattributed[35,49]. Thus, the larger OS concentrations consistently observed through indirect methods indicate that the direct quantification of individual OSs likely underestimates the total OS concentration in ambient OA.

To improve the detection of individual, low-abundant OSs in ambient aerosol samples, we developed a solid-phase extraction (SPE)

method for their enrichment and fractionation. To overcome the need for authentic OS standards, we made use of HPLC coupled to a charged aerosol detector (CAD) alongside a HRMS for separation, quantification and identification, respectively. The CAD is favorable for the quantification of non-volatile species due to its universal response, regardless of the chemical structure[50]. Owing to the terminal sulfate groups (R-OSO₃H), in particular, OSs exhibit a low saturation vapor pressure, thus making them well-suited for quantification using a CAD. In addition to the CAD measurements for quantification, we employed two different mass spectrometric experiments for unambiguous molecular identification: (1) full-scan measurements (fullMS) as a basis for non-target analysis and unambiguous molecular formula attribution of chromatographically well-resolved compounds, and (2) all-ion fragmentation (AIF) experiments for the detection of OS-specific ion fragments across the whole chromatogram. Applied on ambient aerosol filters from different environments, we directly quantified a large fraction of unresolved OSs, indicating their significant presence in both urban and rural environments and, furthermore, highlighting that their contribution to the formation of the aerosol mass cannot be overlooked.

## Results and discussion
### Organosulfate isolation

In the field of atmospheric sciences, water or organic solvents are commonly used for liquid extraction of aerosol filter samples. The application of SPE for enrichment and fractionation of compounds is rarely practiced, but this sample preparation enables novel analytical approaches. Here, we used a mixed-mode anion-exchange (MAX) cartridge and reversed-phase sorbent that has a strong affinity to bind OSs, thus allowing the isolation of acidic compounds from the complex sample matrix of ambient aerosol filters[39]. A fundamental property of OSs, the terminal R-OSO₃H group, leads to a higher acidity (pKa ~ −2.4 to −4.6) for most of the (nitrooxy-)OSs compared to other acidic organic compounds in ambient air (e.g., nitro-phenols or organic acids), ensuring their almost complete dissociation under typical aerosol pH conditions (1–4)[51].

We extracted $PM_{2.5}$ filter samples from Handan, China, and the Taunus Observatory (TO), Germany, representing the North China Plain with severe air pollution, and a relatively clean, rural background station in central Europe, respectively. In the following, we first describe a representative sample from Handan with regard to our new enrichment and quantification approach, before discussing all analyzed samples from both field sites. We have compared the chemical fingerprint with retention time (RT) vs. the mass-to-charge ratio (m/z) of an aliquot of the native aerosol filter extract against the two isolated fractions and the flowthrough after SPE (Fig. 1). The two SPE fractions are diluted to the same concentration level as the native extract. This enabled a quantitative comparison and the evaluation of the absolute SPE-recovery of the single organic compounds detected by non-target analysis. Most of the identified organic compounds in the native extract were found to be smaller than 350 Da and eluted over the entire retention time range (Fig. 1a). CHNO compounds (containing carbon, hydrogen, nitrogen, and oxygen) were the predominant compound class, followed by CHO compounds. The sulfur-containing compounds (CHOS, CHNOS) contributed 37% (by intensity) to the overall detected compounds (Figs. 1a, 2). By using a weak acid dissolved in methanol for the first SPE elution, we removed the less acidic compounds, including (nitro-)aromatics and organic acids (Fig. 1b). Subsequently, we used diluted hydrochloric acid dissolved in methanol to elute the OSs from the cartridge. We did not observe any evidence for degradation of the OSs, confirming their chemical stability at low pH[52]. With this increased solvent acidity for the second SPE elution, we were able to elute the CHOS and CHNOS compounds and, hence, isolate the OSs in a specific fraction. The diluted aliquot of this fraction will now be referred to as the diluted OS fraction (Fig. 1c). The compounds in the flowthrough

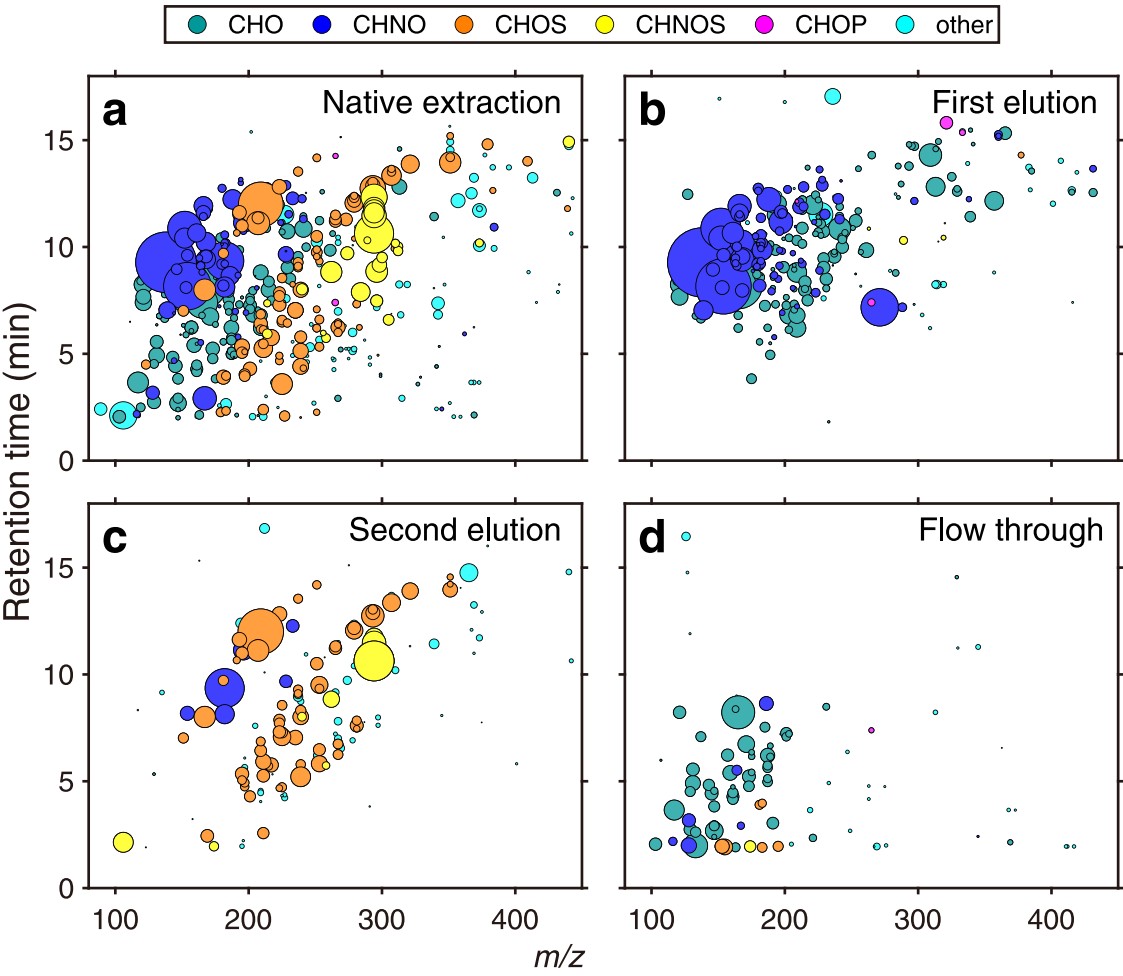

**Fig. 1 | Particulate Matter (PM) sample fractionation by solid-phase extraction.** Mass-to-charge ratio ($m/z$) vs. retention time space showing single organic compounds extracted from a representative Handan ambient $PM_{2.5}$ filter (Handan 1: 18.10.2018, daytime) with negative electrospray ionization. Each circle represents a detected compound with the circle size illustrating the signal intensity. The six compound classes are classified by color (CHO, dark cyan; CHNO, dark blue; CHOS, orange; CHNOS, yellow; CHOP, pink; other, light blue). The compound class names refer to the elemental composition of the classes (i.e., CHNOS must contain the elements carbon, hydrogen, nitrogen, oxygen and sulfur); "other" refers to the compounds excluded from the major compound classes mentioned above. Graph **a** shows the native extraction: ambient $PM_{2.5}$ filter extracted with a water/methanol mixture (98/2 ($v/v$)), **b** the first solid phase extraction (SPE) elution: fraction containing (nitro-)aromatics and organic acids, **c** the second SPE elution: isolated fraction containing organosulfates (OSs) and nitrooxy-OSs, and **d** the flowthrough: native extraction after passing through the SPE cartridge, containing mostly neutral and polar compounds.

were found to be mainly neutral compounds that were not retained by the MAX cartridge (Fig. 1d).

Within the diluted OS fraction, the $MS^2$ spectra revealed that over 99% of the CHOS and CHNOS compounds (by signal intensity) formed a fragment at $m/z$ 96.9601 ($HSO_4^-$), indicating the existence of R-$OSO_3$H groups[53]. Furthermore, 72% of the compounds in the OS fraction had an absolute recovery above 50% compared to the native extraction. Some OSs had a poorer recovery of between 10 and 50%, showing relatively small signal intensities (Fig. S1a). The low recovery of some compounds can be explained by their strong retention on the ion exchange cartridge. By extracting one sample via SPE in triplicate, we found that the intensity-weighted average of the relative standard deviation (RSD) of the SPE recovery was ~15% for OSs, with larger uncertainties for the low-intensity compounds (Fig. S1b). The recovered OSs spanned a large range regarding the polarity and mass-to-charge ratio, ranging from relatively small (~150 Da) and polar (short RT) to large (~350 Da) and non-polar compounds (long RT). However, we did note that the used SPE cartridge provides a low recovery of small and highly polar OSs, which may include certain isoprene-derived OSs, an important class of atmospheric OSs. To a lesser extent, reversed-phase LC (used in this study) is also less

effective for separating these compounds, highlighting the complementary role of hydrophilic interaction LC methods in analyzing smaller OSs[54].

With our SPE method, we obtained an overall mean OS (CHOS and CHNOS) recovery of 75% (with a standard deviation of 7%) for the six ambient aerosol samples (area-weighted mean, Fig. 2). We calculated the overall mean recovery by calculating the sum of all isolated OSs, instead of averaging the recovery of individual OSs. In particular, the mean overall recovery for CHOS-only OSs was found to be 89%. Distinguished by the sampling site, the recovery of all OSs (CHOS and CHNOS) for the Handan samples was 73%, with the CHOS-OS recovery being 91%. The OS recovery for the TO samples was 76%, with the recovery of CHOS-OSs also being 76%. The difference in CHOS-OS recovery for the two sites was likely caused by different OSs with different properties and total aerosol mass loadings. However, we note that larger, less polar OS compounds may exhibit a reduced extraction efficiency, which should be considered in future studies. Overall, the SPE isolation method was found to be capable of effectively isolating the OSs from complex aerosol filter extracts, thereby allowing the subsequent chemical analysis of the enriched fraction and the quantification of the OSs by the CAD.

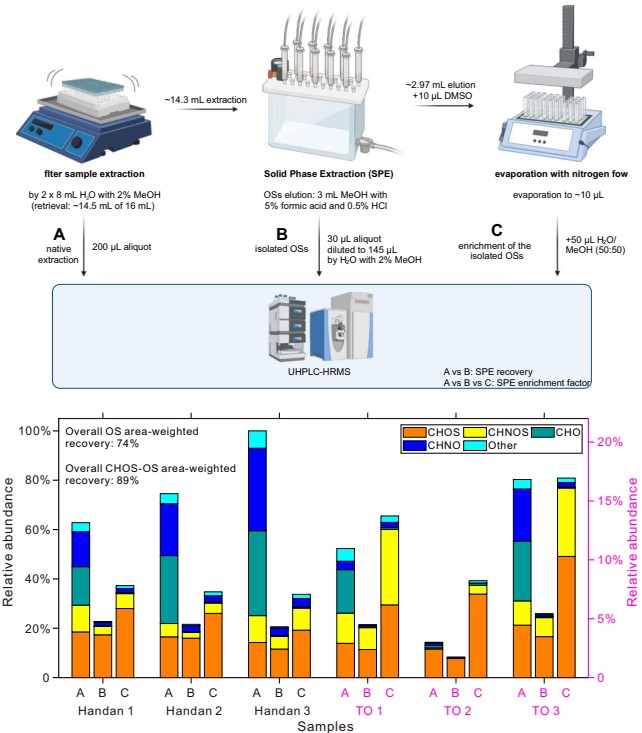

**Fig. 2 | Schematic of the extraction and enrichment procedure. Relative abundance (%) of summed peak areas (negative electrospray ionization) of the different compound classes in the native extraction (A), diluted organosulfate (OS) fraction (B) and enriched OS fraction (C).** The y-axis shows the relative abundance of the compound classes, normalized to the most intense sample (native extraction of the Handan 3 sample). For better clarity, the left y-axis refers to the Handan samples (black; 0–100%), and the right y-axis to the Taunus Observatory (TO) samples (pink; 0–20%). The samples used were Handan 1: 18.10.2018, daytime; Handan 2: 21.10.2018, daytime; Handan 3: 21.10.2018, night-time; TO 1: 11.12.2021, daytime; TO 2: 14.01.2022, night-time, and TO 3: 28.02.2022, night-time. In order to make the native extraction (A), the diluted OS fraction (B) and the enriched OS fraction (C) comparable, we accounted for the enrichment factor by the SPE (~240) and the different experimental setup in (C): (A) and (B) were acquired using HPLC-HRMS (100% of mobile phase flow into the MS) and (C) by using HPLC-HRMS/CAD (16.5% of mobile phase flow into the MS). The consistently larger relative abundance of OSs in fraction (C), therefore, solely originates from additionally detected OSs following SPE. Created in BioRender. Reininger, N. (2025) https://BioRender.com/n70h222.

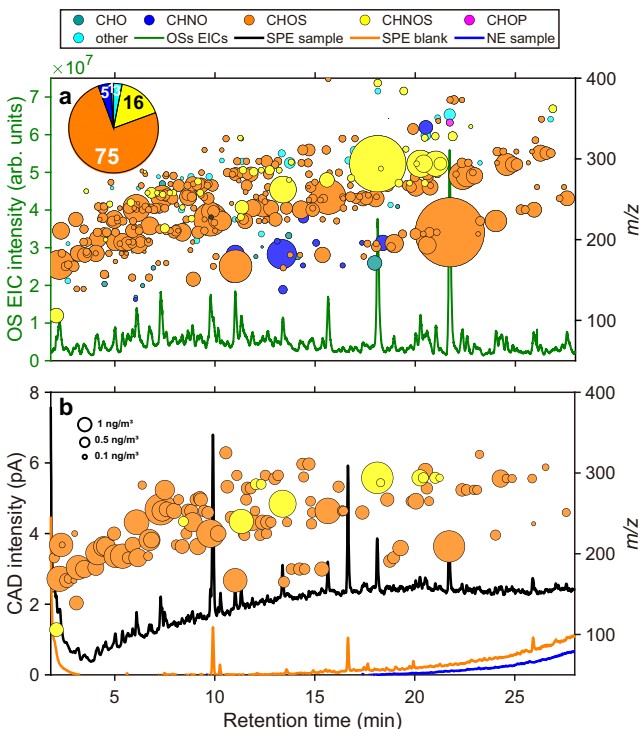

**Fig. 3 | Comparison of the chromatograms and fingerprints of the enriched organosulfate fraction of a representative Handan filter (Handan 1: 18.10.2018, daytime).** The x-axis, displaying the retention time (RT (min)), is zoomed into 2.5-28 min of a total method duration of 40 min. There are no peaks outside this time window. The complete CAD chromatogram is shown in Fig. S9. Graph **a** shows the UHPLC-HRMS data of the enriched OS fraction: RT-*m/z* plot (circles, right y-axis) and the sum of the extracted ion chromatograms (EIC, dark green line, left y-axis) of all the OSs appearing in the RT-*m/z* plot. The pie chart illustrates the peak area contribution of the different compound classes to the total detected compound area. Graph **b** shows the comparison of the UHPLC-CAD chromatograms and the quantified concentrations of individually identified OSs (circles). The three chromatograms show the enriched OS fraction of the Handan 1 filter sample (SPE sample, black line), the corresponding procedural blank of a field filter blank (SPE blank, orange line), and the native extraction (regular liquid extraction technique) of the same ambient filter sample (NE sample, blue line). The circle size represents the organosulfate concentration, determined from the calibration surface (Fig. S4).

## Organosulfate enrichment and quantification

In the enriched OS fraction (second SPE elution after evaporation and reconstitution, see Fig. 2–workflow (C)), the OSs were enriched by a factor of ~240 compared to the native extraction, which thus enables their detection by the less sensitive CAD. This large enrichment factor also increased the number of detected OSs after the non-target analysis of HRMS data (diluted fraction *vs.* enriched fraction of sample "Handan 1": 79 and 490 detected OSs, respectively; see Fig. 1c *vs.* Fig. 3a) due to the improved signal-to-noise ratio of low-abundant OSs. The molecular fingerprints of the enriched OS fraction of all samples are shown in Fig. S2. After SPE, the larger number of OSs revealed the specific molecular characteristics for both field sites, especially for the aliphatic OSs—likely to be of anthropogenic origin—that were clearly visible at the rural TO station during winter time[22,55]. For example, sample TO 2 (Fig. 2), a PM2.5 sample obtained during relatively clean conditions, showed the largest increase in area-weighted CHOS compounds after enrichment. Note that the increase in relative abundance from TO 2-A to TO 2-C (Fig. 2) originates from the larger number of detected OSs following SPE-enrichment. The signal intensity of the enriched OS fraction was 2.8–3.4 times higher than the diluted OS

fraction in the TO samples. For the Handan samples this factor ranged from 1.2 to 1.6 (Fig. 2). With our regular liquid extraction method (i.e., Fig. 1a, native extraction) many OSs remained below the HRMS detection limit, leading to a potential underestimation of the OS contribution to total OA, especially for rural stations.

Figure 3b shows the differences of the CAD chromatograms obtained for the native extract and the enriched OS fraction of the same sample (Handan 18.10.2018, daytime), and the corresponding aerosol filter blank that experienced the full SPE procedure. These chromatograms clearly demonstrate the necessity of the enrichment by SPE for quantification with CAD (Fig. 3b). Our regular liquid extraction of the ambient aerosol filters yielded no detectable peaks in the CAD, whereas strong signals appeared in the OS fractions following SPE. The two highest peaks in the CAD chromatograms (RT ~ 9.91 and ~16.66 min) appear in both the ambient aerosol samples and the blanks following SPE. The parallel detection by HRMS showed no signal of OSs, thus indicating background signals originated from either the SPE cartridge, HPLC column, or aerosol filter extraction procedures. Apart from these two peaks, the majority of the CAD peaks lay within a three seconds retention time tolerance within the summed extracted ion chromatograms (EICs) of the OSs, after correcting for the retention time offset between both detectors. Furthermore, the two largest

CHNO-signals ($m/z$ 182.01, RT ~ 11.01 and ~13.34 min; Fig. 3a) did not result in prominent CAD peaks (Fig. 3b), indicating that their ionization efficiency in HRMS was either very large, or that they were not detected by CAD due to volatilization. The two signals were tentatively identified as 3-nitrosalicylic acid (RT ~ 11.01 min) and 5-nitrosalicylic acid (RT ~ 13.34 min), which have been reported as markers for biomass burning[56,57]. From a chemical perspective, it is reasonable that these organic acids appear in the OS fraction, as the electron-withdrawing mesomeric effect (−M-effect) of the nitro group stabilizes the negative charge of the conjugated base of the carboxylic acid group (electron delocalization), thereby increasing the acidity of the nitrosalicylic acids.

For quantification of the OSs by CAD, we measured eight OS standards at five different concentrations evenly distributed across the mobile phase gradient. In contrast to (−)ESI-HRMS, the CAD has a universal response to the mass of non-volatile compounds, independent of their chemical structure and functionality (Fig. S3). However, by running an HPLC gradient, the CAD´s sensitivity increases with the increasing organic solvent fraction, which is why we calibrated the CAD with eight OS standards. The resulting calibration surface (Fig. S4) enables quantification of unknown signals across the whole chromatogram. It is worth mentioning that the sample matrix showed no signal suppression in the CAD measurements for all the standards used (Fig. S5), thus justifying the quantification obtained through external calibration. To quantify each individual OS, we subtracted a fitted baseline of the chromatographically unresolved peak from the CAD chromatogram (Fig. S6) and, subsequently, fitted the remaining peaks (Fig. S7b) by referencing the OS EICs (Fig. S7a). Following this approach, we quantified all the chromatographically resolved OSs based on the CAD chromatogram.

By comparing signals from the OSs in the HRMS against the CAD, we observed a strong discrepancy of the ionization efficiency for the different OSs. For instance, the signal of monoterpene-derived nitrooxy-OS $C_{10}H_{17}NO_7S$ (RT 18.1 min and $m/z$ 294.0653; Fig. 3a and Fig. 3b) was ~20 times higher than the smaller nitrooxy-OS $C_6H_{11}NO_7S$ (RT 11.3 min and $m/z$ 240.0183; Fig. 3a and Fig. 3b)[10,16,28,58]. However, their quantified concentrations obtained by CAD were 5.5 ng/m³ and 3.3 ng/m³ (Fig. 3b), respectively. Nevertheless, we found the monoterpene-derived nitrooxy-OS ($C_{10}H_{17}NO_7S$) to be the most abundant OS in the representative Handan aerosol filter sample, thereby indicating the importance of the anthropogenic enhancement of biogenic emissions through OS formation. We identified the highest signal in HRMS (RT 21.7 min and $m/z$ 209.1; Fig. 3a) as anthropogenic octyl sulfate ($C_8H_{18}SO_4$) by comparing the retention time and fragmentation pattern with the authentic standard (Fig. S8), and quantified its concentration to 5.4 ng/m³. A recent study reported octyl sulfate as likely being formed through heterogeneous reactions of octyl hydroperoxide with sulfur dioxide or sulfuric acid[59].

Overall, we were able to quantify ~130 individual OSs in each Handan sample and ~65 OSs in each Taunus Observatory sample (Tables S1–6). The total quantifiable OS concentrations by integration of chromatographically resolved OS peaks for the three Handan samples (18.10.2018, daytime; 21.10.2018, daytime; 21.10.2018, night-time) were 149 ng/m³, 157 ng/m³, and 156 ng/m³, respectively. The similar OS concentration of the three samples indicates relatively constant conditions during the two investigated winter days. Certainly, a full seasonal analysis would provide a more variable picture. In the rural samples at the TO, the OS concentrations ranged from 44–57 ng/m³. It is worth mentioning that the three TO aerosol filter samples used in this study were chosen from a large dataset of ~350 samples, collected from August 2021 to August 2022, due to their previously identified high occurrence of OSs. The difference in concentrations between the two sites indicated, as expected, a much stronger anthropogenic influence on OS formation in the North China Plain, although there was still a significant abundance of OSs at the European rural station.

## Unresolved organosulfates

In five of the six investigated samples, we observed a chromatographically unresolved broad peak over the full range of the CAD chromatogram. We did not identify single OSs from the non-target analysis that can explain this broad chromatographic peak (Fig. 3). This is expected, as the non-target analysis requires sharp chromatographic peaks for their detection. A similar feature of the broad chromatographic peak appears in the CAD chromatograms of all three Handan samples (Fig. 3 and Figs. S10–S11). The CAD chromatograms for the TO samples were less elevated (Figs. S12–S14), but an unresolved broad peak was still significantly different from the field aerosol filter blank that experienced the whole sample preparation procedure. Our explanation for the broad peak in the CAD is that it was caused by chromatographically unresolved OSs that remained hidden after mass spectrometry-based non-target analysis. Possibly, a large number of low-abundant OSs contributed in sum to the CAD signal. To confirm the relationship between the broad peak in the CAD and the potentially hidden OSs, we performed an all-ion fragmentation (AIF) experiment. In this setup of the mass spectrometer, all formed ions are guided through the quadrupole into the collision cell where all ions are fragmented simultaneously. Specific ion fragments from OSs ($HSO_4^-$, $HSO_3^-$, $SO_4^{\bullet-}$ and $SO_3^{\bullet-}$) are clearly visible in the Handan samples. We found that the sum of these four ions produced a similar broad peak over the whole AIF chromatogram compared to the CAD chromatogram (Fig. 4a, b). Therefore, we interpreted the elevated baseline in the AIF sample chromatogram to be caused by unresolved and low-abundant OSs in the sample. The blank measurement (with internal OS standards, numbers 1–8, Fig. 4b) did not result in an elevated baseline using the AIF mode. To investigate the correlation between the CAD and AIF measurements, we averaged the two chromatograms in three second intervals to plot the blank-subtracted scatter diagram (Fig. S15). In Fig. S15, the data points appear relatively widely scattered (R = 0.45) with a low RT-dependence. After applying the calibration function on the integrated CAD area (see the polynomial function, Fig. S4), we found a clear RT-dependent gradient between the CAD concentration and the signal of OS fragment ions of the AIF mode (R = 0.18). Division of the AIF chromatogram by the methanol content of the mobile phase at each consecutive time step resulted in an improved correlation between the OS concentration and AIF signal (Fig. 4d, R = 0.90). This observation indicates that the ionization efficiency of the OSs is strongly dependent on the mobile phase composition, with a better ionization efficiency of OSs observed at a higher organic content. Therefore, we suggest that further studies can use this correlation between the quantitative OS measurements by CAD and the quotient of the OS AIF signal over the methanol mobile phase fraction, with the goal to quantify OSs via HRMS only. Besides this observation of a novel way to quantify the total content of resolved and unresolved OSs in ambient aerosol, the detection of OS-specific ions by AIF that correlate strongly with the CAD signal supports our hypothesis that the chromatographically unresolved broad peak in the CAD chromatograms was mainly induced by hidden OSs.

For total OS quantification, we fitted the chromatographically unresolved broad peak as one large peak (Fig. 5a). The resulting integration showed that ~2.3, ~2.8, and ~2.4 µg/m³ of OSs were chromatographically unresolved for the Handan 18.10.2018 daytime, 21.10.2018 daytime, and 21.10.2018 night-time samples, respectively. Figure 5b indicates that the chromatographically resolved OSs of the Handan samples contributed ~0.7-1% to the total organic matter (OM), while the overall area of the CAD chromatogram (representing the resolved and unresolved OSs) contributed 12-17% to the OM. Although the TO samples exhibited a lower total OM, we found an overall larger fractional contribution of OSs. The TO 2 (14.01.2022, night-time) sample showed the lowest concentration of OSs, at ~0.04 µg/m³ that were chromatographically resolved. No chromatographically unresolved OSs were visible in this sample (Fig. S13). The contribution of the individually

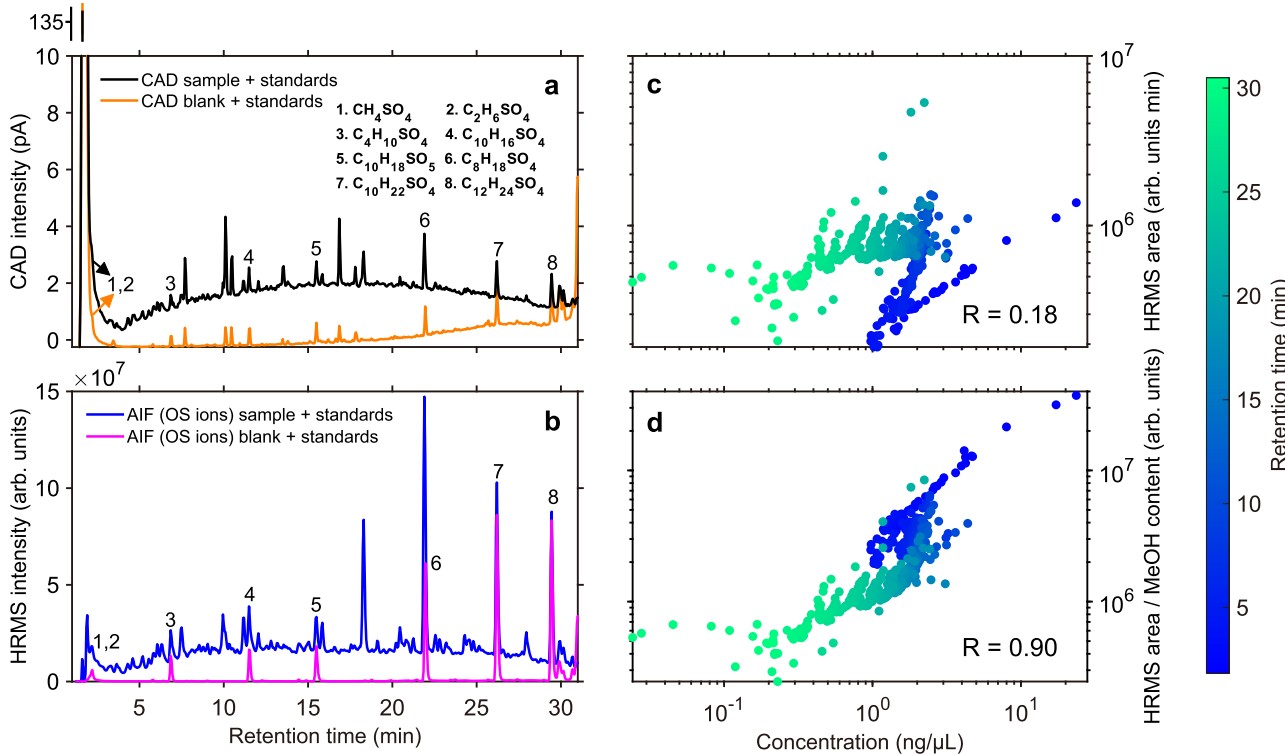

**Fig. 4 | UHPLC-CAD and UHPLC-HRMS chromatograms of the second organosulfate-containing fraction.** The HRMS chromatogram is the sum of the extracted ion chromatograms (EICs) of the organosulfate-specific ions HSO4−, HSO3−, SO4•− and SO3•− recorded in the all ion fragmentation (AIF) mode. This correlation is demonstrated by using results from the Handan 1 aerosol filter. Graph **a** shows the UHPLC-CAD chromatograms of the Handan ambient aerosol sample (black) and the ambient aerosol filter blank (orange), **b** the UHPLC-HRMS chromatograms of the Handan ambient aerosol sample (blue) and the ambient aerosol filter blank (pink), **c** correlation of the blank-subtracted OS concentration (determined by the external calibration of the UHPLC-CAD) and the UHPLC-HRMS signal of the OS-specific ions recorded in the AIF mode, and (**d**) same as (**c**), but with the mass spectrometric signal being divided by the methanol content to correct for the ionization efficiency. The peak numbering in (**a**, **b**) represents the eight spiked internal standards. The color bar of (**c**, **d**) shows the retention time (min).

quantified OSs to the total OM was ~34%, although the uncertainty of this value remains large due to the low concentration of OSs (Fig. 5b). The homologous series of aliphatic OSs in this sample suggests an anthropogenic origin (Fig. S2e). Backward trajectories indicate a possible marine origin from the North Sea (Fig. S16) with aliphatic OSs being emitted by the shipping sector[60]. The concentration of total OSs for TO 11.12.2021 daytime and TO 28.02.2022 night-time was ~0.6 and ~0.8 μg/m³, hence, the total OSs contributed ~21% to the total OM for both samples (Fig. 5b). This large OS contribution to the total OM at TO is certainly an upper estimate since our sample selection for this field site was guided by a pre-evaluation of a full-year compositional analysis. Furthermore, we cannot rule out that a fraction of OSs is formed during aerosol filter sampling, although the SO₂ concentrations at this rural background station are usually below quantifications limits for standard detectors. Hence, we infer that the majority of OSs observed at rural and remote stations forms during atmospheric transport. Analysis of the backward trajectories for TO 3 indicate that the air masses originated from eastern Germany, southern Poland and the northern Czech Republic, which are Europe's hotspots of coal combustion by power plants (Fig. S16)[61]. The time of atmospheric transport from this source area of around one day is twice as long as the actual aerosol filter sampling period, which speaks against significant OS artifact formation for the TO aerosol filter samples.

To evaluate the quantitative contribution of OSs to total OM, we compared the major quantification studies that followed either the direct or the indirect approaches (Fig. 5c and Table S7). The indirect methods determine a larger OS fraction contributing to the OM (on average ~15%), whereas the direct approaches reveal a smaller fraction of OSs in the OM (~2%)[12,28,29,33,35,36,41,43,44,47,49,62–66]. Our method presented

here enables the direct quantification of the hidden OS fraction and, consequently, suggests that the direct quantification of molecularly resolved OSs generally underestimates the total OS content in ambient aerosol samples by roughly a factor of five to ten. Thus, the described quantification approach for estimating OSs using SPE and UHPLC-CAD enables the direct quantification with lower uncertainties compared to the approaches that employ two different methods. Further assumptions, for example, about the average molecular weight of OSs, are no longer necessary.

Our observation of the high contribution of OSs at both locations, in urban China and rural Europe, calls for further investigation of the variability and seasonality of OSs in atmospheric aerosol across the globe. The described observations are in agreement with the indirect quantification methods of organic sulfur species. Due to the enrichment of individual OSs that enable a more detailed OS fingerprint (Fig. S2), we are now able to evaluate the fractions of OSs that are purely anthropogenic (e.g., saturated aliphatic OSs) or whether they can be seen as anthropogenically enhanced OSs (e.g., sulfate from anthropogenic sources with a carbon backbone from biogenic sources). Especially, isoprene-derived OSs might still be underestimated using the described method, as we found the SPE-recovery for these polar compounds is around 30% (Fig. S17). Interestingly, some of the OSs at the rural Taunus Observatory indicate the long-range transport of anthropogenic pollution. The low abundance of terpene-derived OSs can be explained by very low biogenic emissions of the investigated samples from the winter season. Further analysis of year-long aerosol filter analyses should reveal the pure anthropogenic and mixed biogenic/anthropogenic OS quantities, thus enabling a better understanding of pollution over whole seasons at different locations.

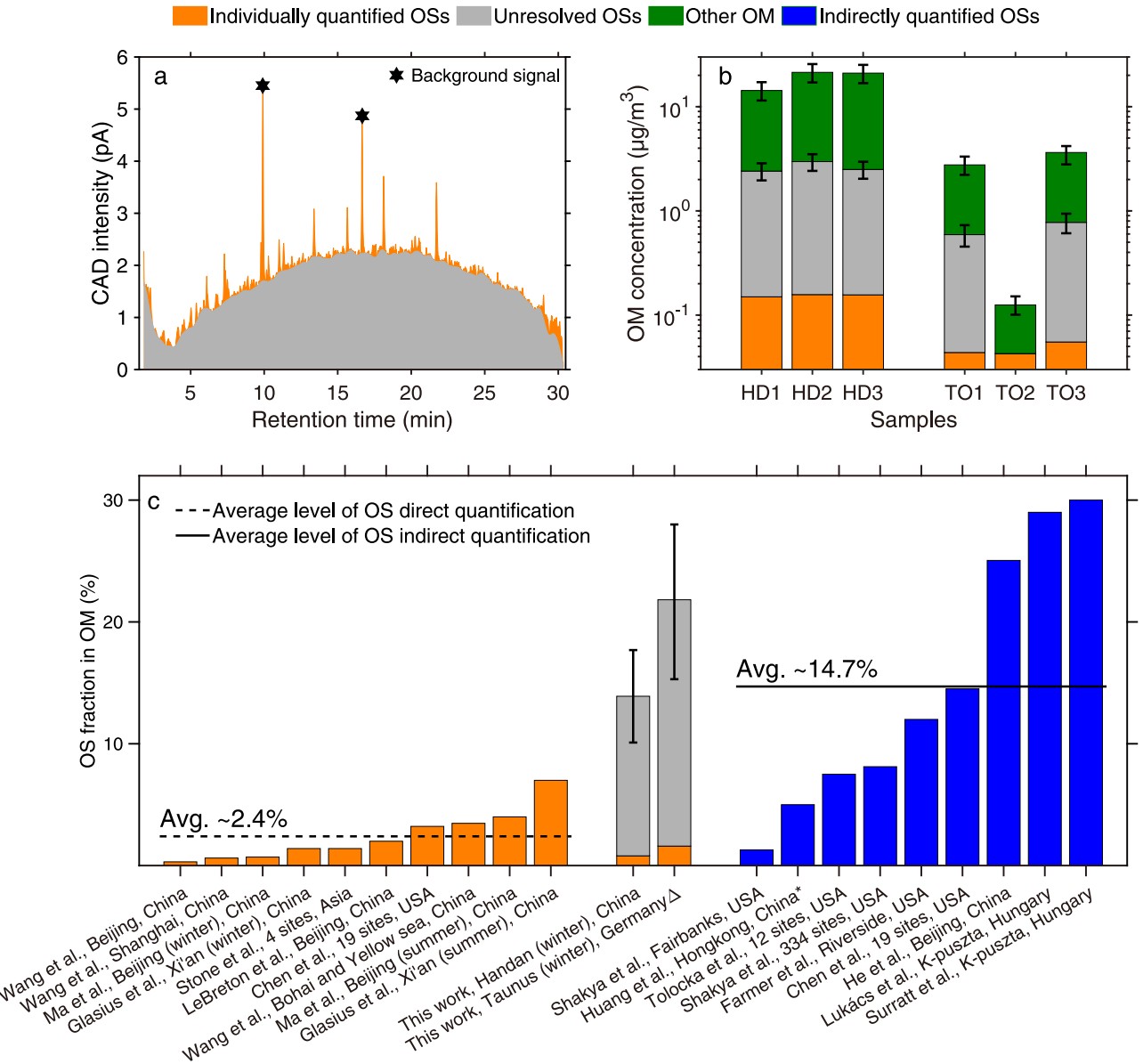

**Fig. 5 | Integration of the chromatographically unresolved peak of the charged aerosol detector (CAD) measurements, its contribution to the total organic matter (OM), and its comparison with other studies.** Graph **a** shows the integration of the CAD chromatogram (Handan 1, 18.10.2018 daytime): gray area and orange peaks show unresolved and resolved organosulfates (OSs), respectively, **b** shows the comparison of the concentration (µg/m³) of resolved OSs (orange), unresolved OSs (gray) and other OM (dark green) for the six investigated samples (Handan 1, 18.10.2018 daytime; Handan 2, 21.10.2018 daytime; Handan 3, 21.10.2018 night-time; Taunus Observatory (TO) 1: 11.12.2021, daytime; TO 2: 14.01.2022, night-time; TO 3: 28.02.2022, night-time). The OM and OSs concentration error bars

represent the uncertainty for the organic carbon-to-organic matter conversion, the SPE reproducibility, and CAD quantification. Graph **c** shows the comparison of the OS fraction obtained in the OM between different studies; directly quantified OS (orange and gray) vs. indirect quantification (blue) (See Table S7). The dashed and solid lines are the averages of the OS fractions from the direct and indirect quantification studies, respectively. Error bars represent the combined error from solid phase extraction reproducibility, CAD quantification, and OC to OM conversion for each site. We note that the OS mass fraction includes the sulfate group. *Minimal estimation, △Only TO 1 and TO 3 being calculated.

## Methods

### Aerosol filter sampling, organic carbon analysis, and solid-phase extraction

Six 12 h integrated PM$_{2.5}$ filter samples were collected at two sampling sites. The first sampling site was located at Hebei University of Engineering (36.57°N, 114.50°E), an urban site in Handan, China. We collected three Handan samples on preheated quartz fiber filters (203.2 mm × 254 mm, QM-A Quartz Microfiber Filters, Whatman™, China) with a high volume sampler (TH-1000 C, Tianhong, China; sampling flow rate 1 m³/min) on 18.10.2018 (daytime, 08.00–19.30), 21.10.2018 (daytime, 08.00–19.30) and 21.10.2018 (night-time,

20.00–07.30). The second sampling site was located at the Taunus Observatory (TO; 50.22°N, 8.45°E, altitude 825 m), a mountaintop site at the Kleiner Feldberg in Hesse, Germany. We collected three TO samples on preheated glass fiber filters (Ø 150 mm, MG 160, Ahlstrom-Munksjö, USA) by using a high volume sampler (DHA-80, Digitel, Switzerland, air volume 500 L/min) on 11.12.2021 (daytime, 07.00–19.00), 14.01.2022 (night-time, 19.00–07.00) and 28.02.2022 (night-time, 19.00–07.00).

We took 1.45 cm² of each aerosol filter and performed the OC analysis by using a thermal/optical carbon analyzer (Model 4, Sunset Lab.). We deployed the NOISH 870 protocol for Handan aerosol filters

(quartz fiber filter) and the Semi-Continuous OC/EC Analyzer with a program featuring a lower temperature setting of up to 500 °C for TO aerosol filters (glass fiber filter). We used an OC-to-OM conversion factor of 1.6 and 1.8 for Handan and TO, with the assumption of 20% relative uncertainty, respectively[12,67].

We sampled a 24.5 cm² and a 13.0 cm² filter punch from each Handan and TO ambient aerosol filter, respectively (we determined the area of the filter punch based on the amount of sampled air volume). We extracted the filter fragments by using an orbital shaker (Edmund Bühler GmbH, Germany) for $2 \times 20$ min, each time using 8 mL of a water/methanol mixture (98/2 (v/v); water: Milli-Q water, 18.2 MΩ·cm; Milli-Q® Reference A+ System with Millipak® Express 40 Filter (0.22 μm), Merck Millipore; methanol: UHPLC-MS grade, Thermo Scientific™). We combined the two consecutive native extractions, centrifuged at 4000 rpm for 40 min, and transferred ~14.5 mL of liquid supernatant to new vials. We saved an aliquot of the native extract for a direct measurement and used the remaining extract for the SPE separation.

After conditioning the SPE cartridge (Oasis MAX, 1cc/10 mg bed weight, Ø 30 μm particles, Waters™, Milford, MA, USA) with 1 mL each of (1) methanol, (2) Milli-Q water, (3) 5% $NH_4OH$ in Milli-Q water, and (4) extraction solution (methanol/water mixture), we loaded the native aerosol filter extracts onto the cartridge and collected the flow through for further analysis. Omitting a drying step, we directly eluted the cartridge twice using (1) 6 mL of 2% formic acid in methanol, followed by (2) 3 mL of 5% formic acid and 0.5% hydrochloric acid in methanol (Fig. S18). The two elutions were collected separately. An aliquot of each elution was taken and diluted with the extraction solution to the same concentration level as the native extract.

We added 10 μL of dimethyl sulfoxide to the second elution as a preservative and to ensure the suspension of compounds before evaporating the sample under a gentle nitrogen flow to a volume of ~10 μL. After the evaporation, we used 50 μL methanol/water mixture (50/50 (v/v)) to re-dissolve the remaining sample for subsequent analysis.

### UHPLC-CAD-HRMS analysis

We measured the native extract, the diluted first SPE elution, diluted second SPE elution and the flow-through via the Orbitrap HRMS (QExactive Focus™ Hybrid Quadrupol-Orbitrap Mass Spectrometer, Thermo Fisher Scientific™) equipped with a UHPLC system (Vanquish Flex UHPLC System, Thermo Fisher Scientific™). We operated a reversed-phase column (CORTECS T3, 120 Å, 2.7 μm, 3 mm × 15 mm, Waters) in the gradient mode (at 40 °C, still air) to achieve chromatographic separation. The eluents were Milli-Q water (eluent A) and methanol (eluent B), both containing 0.1% formic acid (v/v). The mobile phase gradient started with 1% eluent B (0–1 min) which was then increased linearly to 99% B (1–14 min), kept at 99% B (14–15.5 min), and then decreased to 1% B (15.5–16.5 min); the column remained at 1% B (16.5–20 min) for re-equilibration. The total method duration was 20 min with a flow rate of 0.4 mL/min and an injection volume of 4 μL. Ionization was conducted in the negative mode via a heated ESI (HESI-II Probe, Thermo Fisher Scientific™). Data acquisition for m/z was set from 75 to 750, with a resolving power of ~70,000 at m/z 200. The settings for the ion source were as follows: 8 psi auxiliary gas (nitrogen), 40 psi sheath gas (nitrogen), 3.5 kV spray voltage, and 350 °C gas temperature.

CAD has a sensitivity in the low-nanogram to high-microgram range (on column), which allows for the detection of trace levels of non-volatile compounds. Hence, we measured the concentrated second SPE elution using UHPLC-CAD/HRMS. The LC method was adjusted by increasing the column temperature to 50 °C and prolonging the total method duration to separate the detected compounds further. The mobile phase gradient started with 1% eluent B (0–1 min) increasing linearly to 99% B (1–34.5 min), and was kept at 99% B (34.5–36.5 min) and then decreased to 1% B (36.5–38 min), and

remained at 1% B (38–40 min) to re-equilibrate the column. The total method duration was 40 min. We utilized a post-column flow splitter (Quicksplit 610, Analytical Scientific Instruments US) to direct 83.5% of the mobile phase flow (0.4 mL/min) into the CAD (0.334 mL/min) and the remaining flow into the MS (16.5%, 0.066 mL/min). Due to the split of the flow rate, the auxiliary and sheath gas pressures of the ion source were changed to 10 and 31.5 psi, respectively. The settings for the CAD were as follows: 25 °C evaporator temperature, 2 Hz data collection rate and 1.00 power function value. We subsequently measured the enriched second SPE elution in the all-ion fragmentation (AIF) mode via the UHPLC-HRMS with the same gradient as in the CAD measurement. AIF acquisitions were performed with a resolving power of ~70000 at an m/z ratio of 200, using a stepped collisional energy of 30, 40, and 55 units, and an automatic gain control (AGC) target of $3 \times 10^6$.

### Data analysis

The non-target analysis software Compound Discoverer (CD; version 3.3, Thermo Fisher Scientific™) was used to identify the chromatographic peaks. By comparing the exact mass, isotopic matching score and MS/MS fragmentation pattern with the database mzCloud™ (Thermo Fisher Scientific™), molecular formulae could be assigned to the detected peaks. The detailed settings of the CD-workflow for the UHPLC-HRMS and UHPLC-CAD/HRMS sequences are provided in Tables S8 and S9.

In brief, a threshold intensity of $1 \times 10^4$ and a peak rating of larger than five was applied to the two-dimensional coordinate system (UHPLC-HRMS sequence: y-axis RT 0–20 min, x-axis m/z 50–750; UHPLC-CAD/HRMS sequence: y-axis RT 0–40 min, x-axis m/z 50–750) for all measurements. Only chromatographic peaks exceeding these thresholds were analyzed. In addition, the maximum peak signal of a compound had to be three times higher than its signal in the blank sample to be considered in the ambient aerosol samples. Molecular formulae were calculated with the permitted elemental combinations of $C_{n1}H_{n2}Br_{n3}Cl_{n4}N_{n5}O_{n6}P_{n7}S_{n8}$ (n1 = 1–90, n2 = 1–190, n3 = 0–3, n4 = 0–4, n5 = 0–4, n6 = 0–20, n7 = 0–1, and n8 = 0–3) and with a mass tolerance of 2 ppm. We set the minimum and maximum H to C ratios as 0.1 and 4.5, respectively, to filter out the chemically unrealistic compounds. We used the software Fityk to fit the individual CAD peaks based on the RT of the EIC of the MS data. CAD quantification of the total OSs was obtained using 95% confidence intervals of the fit. The obtained data from the non-target analysis were further processed and plotted by using self-written scripts using Matlab (Matlab R2023b, Mathworks).

## Data availability

All data used to generate figures are available under https://zenodo.org/records/14609502[68].

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

## Acknowledgements

This research has been supported by the Deutsche Forschungsgemeinschaft (DFG; German Research Foundation, grant nos. 410009325 and 428312742 (TRR 301)) (A.L.V., T.H.). This work received funding from the Robust-Nature Cluster of Excellence Initiative provided by the Goethe University Frankfurt, Germany (A.L.V.). Jialiang Ma thanks the Open Foundation of Hebei Key Laboratory of Resource Survey and Research and the life expense covering from China Scholarship Council (CSC) (NO.202008080054) (J.M.). Cunliang Zhao acknowledges support by Science and Technology Project of Hebei Education Department (JZX2023010) (C.Z.). Yuchen Wang acknowledges the following projects: the National Nature Science Foundation of China (22306059) (Y.W.), the Science and Technology Planning Project of Hunan Province (Grants 2023JJ40128) (Y.W.), the Nature Science Foundation of Changsha (kq2208019) (Y.W.), and the State Key Laboratory of Loess and Quaternary Geology, Institute of Earth Environment (SKLLQG2235) (Y.W.). Bangjun Liu acknowledges Funding Project for the Introduction of Overseas Returnees in Hebei Province (No. C20230365) (B.L.). Hongya Niu thank the Central Guide Local Fund for Scientific and Technological Development of Hebei Province (236Z3702G) (H.N.).

## Author contributions

J.M. did the data curation and analysis, and wrote the original manuscript. J.M. and N.R. did the SPE experiment. J.M. performed the UHPLC/CAD/HRMS measurements and nontarget analysis. C.Z., B.L., and H.N. collected the ambient filters from Handan. D.D., J.R., Y.Q., and Z.W. did the OC quantification experiment. A.B., J.D., S.H., and T.H. performed the oxidation flow reactor and HILIC experiment. Y.B., Y.L., Y.X., L.L., and Y.W. synthesized and purified the 2-hydroxy-α-pinene organosulfate. A.L.V., N.R., F.U., M.S., and L.D. edited and revised the manuscript. A.L.V. managed the project and supervised the experiment. All authors have contributed to the scientific discussion and have given approval to the final version of the manuscript.

## Funding

## Competing interests

The authors declare no competing interests.
