## [Transparent Peer Review file · Nature Communications]

Unveiling a large fraction of hidden organosulfates in ambient organic aerosol

Corresponding Author: Professor Alexander Vogel

Version 1:

Reviewer comments:

Reviewer #1

(Remarks to the Author)

The manuscript by Ma et al. presents interesting, novel results regarding organosulfates in atmospheric aerosol particles. This group of compounds were discovered in aerosols less than two decades ago and their formation and occurrence have been studied in both laboratory and ambient settings. However there has been a discrepancy between the concentrations in ambient air inferred from unspecific total sulfur aerosol measurements and concentrations estimated based on chromatographically resolved individual compounds detected by mass spectrometry (MS). The present study bridges this gap in our understanding through development of another type of detection method for organosulfates by application of a so-called Charged Aerosol Detector (CAD) which enables quantification of the total mass in a chromatographic peak, without being affected by differences in ionization efficiency as with MS detectors. Furthermore the authors are able to detect a broad peak of unresolved organosulfates in the chromatograms. Together these observations enable the authors to conclude that their aerosol samples contain a much higher concentration of organosulfates than detected with standard chromatography-MS methods, which is closer to the observations from total sulfur measurements.

The study will be of broad interest to the scientific community, as it points out that organosulfates have a much larger prevalence in atmospheric aerosols than generally thought. I expect that the study will lead to many new investigations on formation and occurrence of organosulfates in the atmosphere.

In general the study is well presented. However the manuscript needs to be checked by a professional who is a native English speaker. One example of issues that needs to be corrected, is the wrong use of the word "exemplary" (e.g. Fig. 3 figure text). As there are several more instances of not grammatically correct English, the manuscript needs to be checked and corrected.

Specific comments

Line 15: ambient samples -> ambient aerosol samples

L17: I suggest to add ", respectively" after organosulfates

L18-19: These sentences may not be clear to a more general reader.

L25: Up to 90% OA in atmospheric aerosols seems very high. Maybe it is more relevant to state a more general value.

L37: I was a bit surprised to see the paper by Le Breton et al. highlighted as the first paper presented in detail here, given that there are a number of earlier studies using UHPLC-MS as in the current work.

L46-47: The use of camphor-10-sulfonic acid seems outdated, especially as the functional group is not a sulfate.

L76-77: These first sentences are unclear. The general reader may not understand that the filter extraction procedure is similar, but sample preparation differs.

L79-80: How does the sulfate group change acidity of nitrooxy-OSs?

L83: Exemplary has a different meaning. Please correct. You probably mean that the sample is used as an example.

L86-87: This sentence is not clear.

L104: How large? Please be specific.

L117: Can you state a standard deviation on the recovery?

L123: Often it is necessary to add the word "aerosol" when you talk about filter extracts to make the sentence more precise.

L140: What is the sensitivity of the CAD? Can this be described briefly, maybe in the experimental section?

L152: It seems strange to refer to Figure 3b before 3a.

L145: Are other data available to support the interpretation? Can TO really be described as remote (which is typically further away from anthropogenic sources)?

L155: ambient filters -> suggest to change to: ambient aerosol filters - or similar.

L164: What is the M-effect?

L190: Two days can hardly be called a period.

L192: It is very unclear to the reader what this means. How were the sample selected?

L194: A verb seems to be missing here. It is important to point out that these were three selected samples during winter, where previous studies have observed relatively high OS concentrations.

L249- and Fig. 5: For the calculation of OM from organosulfates, was the sulfate group omitted?

L256: The North Sea is quite far away from TO compared to the closer densely populated and industrialized areas in Germany and Netherlands. The statement about shipping origin thus seems speculative.

L259: Again the reader is wondering how these three samples were selected.

L261: This has previously been observed by e.g. Glasius et al., Composition and sources of carbonaceous aerosols in Northern Europe during winter, Atmospheric Environment, 173, 2018, pages 127-141.

L278: This seems like a very strong statement for apparently unpublished data.

Figure 1: I suggest to make larger dots in the legend, so they are easier to see.

Figure 2: In the description of the aerosol filter sample extraction in the figure, I suggest to list the composition of the extraction solvent.

Figure 3: Is CHOP a common group of compounds in aerosols?

Reviewer #2

(Remarks to the Author)

Summary and Recommendation:

This manuscript examines the types and amounts of organosulfates (OSs) found in PM_{2.5} samples collected from Handan, China (urban) and Hesse, Germany (mountain top site that is remote) by using solid phase extraction (SPE) as a sample preparation step before RPLC-CAD-HRMS analysis. Although SPE has been rarely used in aerosol chemistry research, it helps in aiding the identification and quantification of OSs when using RPLC-CAD-HRMS. I should note that SPE was originally used in earlier work that analyzed PM_{2.5} samples collected from the SE USA (e.g., Gao et al., 2006, JGR-Atmospheres), but it was later neglected due to it removing small water-soluble OSs (such as the isoprene-derived OSs) in one of the washing steps. This latter recognition was so important to isoprene-derived SOA because using SPE caused these OSs to go initially unrecognized. With that said, to my knowledge, this study uses RPLC coupled with both CAD with ESI-HRMS detection. The CAD offers incredible potential to quantify OSs without the need for all of the authentic standards being available. In atmospheric chemistry, the lack of available authentic standards has been a huge hindrance, preventing models from being updated with the latest multiphase chemical processes that lead to low-volatility OSs. Overall, I find this manuscript exciting and mostly well-written, especially for atmospheric and aerosol chemists. Some readers not well-versed in analytical chemistry might find this paper somewhat difficult to read. However, I hope the Editor won't find this a reason to reject the paper. I do think the demonstration of a large un-resolved OS peak in the chromatogram is notable and appears to contribute a large OA mass to PM_{2.5} collected during the winter seasons in China and Germany. I have several major comments below that should be carefully addressed by the authors before publication in Nature Communications can be fully considered. Due to the nature of my comments below, I recommend the manuscript be reconsidered after major revisions.

Major Comments:

1.) Filter collection artifacts:

There has been some important work to recently come out of Hartmut Hermann's lab (and I believe Marianne Glasius' lab) showing that filter sampling (especially with media like quartz fiber filters) is prone to positive sampling artifacts that lead to the production of certain OSs like m/z 294 (one of the nitrooxy OSs). Since high-volume filter samplers cannot use denuders (due to the high flowrates used) to remove gas-phase SO₂ and NO_x, it is possible that NO_x and SO₂ can absorb from the gas phase and react on these filter types yielding unwanted reactions that lead to OSs. Knowledgeable readers might wonder how much of the mass you quantify is related to these positive sampling artifacts?

2.) Lines 29-31: The authors should cite the earliest papers here for OSs being discovered as an important subclass of OA, including the following:

Romero and Oehme (2005, J. of Atmos. Chem.) - DOI: 10.1007/s10874-005-0594-y

Surratt et al. (2007, ES&T) - DOI: 10.1021/es062081q

linuma et al. (2007, Atmos. Environ.) - <https://doi.org/10.1016/j.atmosenv.2007.03.007>

linuma et al. (2007, ES&T) - <https://doi.org/10.1021/es070938t>

Surratt et al. (2008, JPCA) - <https://doi.org/10.1021/jp802310p>

The Surratt et al. (2008, JPCA) study estimated that upwards of 30% of particulate OM in Hungary could be OSs.

Additional work by Betsy Stone's group has also estimated OS contributions to OM, such as <https://doi.org/10.5194/acp-19-3191-201>. In that study, they estimated 16.5% of PM_{2.5} OC as OSs.

3.) Lines 31-32: I'm a bit surprised the authors didn't cite any work on the reactive uptake of isoprene epoxydiols (IEPOX)

yielding OSs in PM_{2.5}. This is one of the major semi-volatile oxidation products out there that has helped clue us in as a research community on the role of reactive uptake (multiphase chemistry) onto wet acidic sulfate aerosols that yield OSs.

4.) Lines 32-33: Similar to earlier lines in the introduction, simply citing review articles for statements like this doesn't seem appropriate. I think the authors should be citing some of the original studies that demonstrated the role of SO₂ emissions in enhancing SOA from BVOC oxidations.

5.) Lines 36-49: This paragraph is very European- and Asian-centric. I think this paragraph is fine, but a lot of prior OS work has also been done in the USA trying to quantify OSs in PM_{2.5} there. There has been a lot of work by the Surratt Group (UNC) and Stone Group (University of Iowa) on this topic using LC/MS methods and synthesizing authentic compounds. Furthermore, Jian Yu's Group (HKUST) in Hong Kong has also estimated OS contributions in PM_{2.5} and has synthesized authentic compounds as well.

6.) Lines 92-93: How comfortable are the authors assuming that the majority of OSs are stable in the presence of a strong acid? Work from Matt Elrod's group (Oberlin College) has suggested that the stability of OSs depends on whether the OS is primary or tertiary. Did the authors consider testing their approach with different OS standards to confirm the stability of OS compounds?

7.) Figure 1d:

As shown in Figure 1d, there are still some OS compounds that elute through what they call "flow-through." Are these accounted for? One thing I want to note to the authors is that in Gao, Surratt, et al. (2006, JGR-Atmospheres), they employed an SPE separation to measure SOA. When they did this OSs were missed, especially the very water-soluble types (like the isoprene-derived OSs). Later Surratt et al. (2007, ES&T) found that by not using SPE, but rather a polar-end capped C18 column, they could measure the isoprene-derived OSs along with the larger monoterpene-derived OSs. Later, Betsy Stone's group (University of Iowa) and later Surratt's group (UNC) showed that HILIC separation is much more effective in resolving and quantifying isoprene-derived OSs and other water-soluble and smaller OSs. I'm curious, do the authors worry they are still missing water-soluble OSs that are not well retained on C18 columns? I noted the authors conducted reverse-phase liquid chromatography (RPLC) to separate and measure particulate OSs. However, it might be better for all of us in the community to use both HILIC and RPLC separations as orthogonal methods to better separate, characterize and quantify the OSs that range from water-soluble to less water-soluble. Do you agree? It might be worth mentioning the limitations of RPLC in past work and why HILIC methods have come about for the smaller OSs.

8.) Line 117-118: How do the authors know they extracted all of the OS compounds and mass from their filters using 98% water and 2% methanol by volume solvent only? Were standards of OSs also examined through their extraction procedures? I think this would be helpful to include. Could larger OSs be missed if not soluble in water? Also, with water, do you have to worry about the hydrolysis of OSs?

9.) Lines 121-122: I agree that the recoveries can be affected by what the authors state here. But what about also the matrix of the sample? Could that affect the recovery from the filters? For example, it is known that particulate PAHs can be hard to extract from certain filters depending on the matrix of the filter and the other components collected on the filter.

10.) Lines 144-145: The authors note that there are OSs that are aliphatic and of anthropogenic origin. Are the authors aware of work by Riva et al. (2016, ACP)? This study demonstrated OSs form from the oxidation of alkanes in the presence of wet acidic sulfate aerosols. Riva et al. (2015, ES&T) also demonstrated OSs form from PAH oxidations in the presence of sulfate aerosols. Notably, in this latter study, they also observed sulfonates. Did you by chance observe any sulfonates in your PM_{2.5}, possibly from the wintertime samples? What about other oxidation states of sulfur?

11.) Minus the comment I made about the filter sampling artifacts above, I wonder if this method would show even higher OS fractions in fine particulate OM during the summer seasons when BVOC-derived oxidation products are known to yield substantial amounts of OS in the presence of sulfate? The fact the authors show fairly high amounts in winter, tells me the mass fractions could even be higher than previously thought in summer....

12.) Some readers or reviewers might ask why the authors didn't try this RPLC-DAD-HRMS method on a well-characterized lab-generated SOA type like IEPOX SOA before analyzing field samples? The reason I raise this point is for IEPOX SOA our community has an analytical method (i.e., HILIC/ESI-HR-QTOFMS from both the Stone and Surratt Groups) and authentic standards (from the Surratt Group) to compare your quantitative method with. Furthermore, lab studies typically can denude unwanted gases (like SO₂/NO_x) before collecting onto Teflon filters (which are less prone to positive sampling artifacts like quartz).

Version 2:

Reviewer comments:

Reviewer #1

(Remarks to the Author)

The authors have done a very good job addressing the comments from the reviewers. However I have a few additional comments to the response.

Line 18-19

"We found a significantly larger organosulfate fraction appearing as a broad chromatographic peak in the charged aerosol detector." You do not find peaks inside the charged aerosol detector, but maybe in the chromatograms from it. Please use correct wording.

Regarding the investigation of responses of different OS standards, it can be discussed whether the first compound formaldehyde sodium bisulfite has a sufficient response or it should be excluded from the calculation of average response and variation, when evaluating the suitability of camphor-sulphonic acid as reference standard.

"The application of SPE for enrichment and fractionation of compounds is rarely practiced, but this sample preparation enables unique and novel analytical approaches." As the other reviewer pointed out, SPE was already used by Gao et al., 2006, JGR-Atmospheres. Furthermore SPE is used to extract HULIS in aerosol samples for quantification and characterization (e.g. Baduel, C., Voisin, D., and Jaffrezo, J. L.: Comparison of analytical methods for Humic Like Substances (HULIS) measurements in atmospheric particles, Atmos. Chem. Phys., 9, 5949–5962, <https://doi.org/10.5194/acp-9-5949-2009>, 2009.). Based on these considerations, I suggest removing "unique" here.

OS/OM. It is important to clarify in the text, that here the OS mass fraction includes the sulphate group. Meanwhile some of the previous studies listed in Table S7 may have either included or excluded the mass of the sulfate group in the OS fraction. Did the authors check this?

Manuscript ID: NCOMMS-24-43312A-Z

Manuscript Type: Article

Collection: Mass spectrometry method development

Response to reviewers

Reviewers' Comments to the Authors:

Reviewer 1

The manuscript by Ma et al. presents interesting, novel results regarding organosulfates in atmospheric aerosol particles. This group of compounds were discovered in aerosols less than two decades ago and their formation and occurrence have been studied in both laboratory and ambient settings. However there has been a discrepancy between the concentrations in ambient air inferred from unspecific total sulfur aerosol measurements and concentrations estimated based on chromatographically resolved individual compounds detected by mass spectrometry (MS). The present study bridges this gap in our understanding through development of another type of detection method for organosulfates by application of a so-called Charged Aerosol Detector (CAD) which enables quantification of the total mass in a chromatographic peak, without being affected by differences in ionization efficiency as with MS detectors. Furthermore, the authors are able to detect a broad peak of unresolved organosulfates in the chromatograms. Together these observations enable the authors to conclude that their aerosol samples contain a much higher concentration of organosulfates than detected with standard chromatography-MS methods, which is closer to the observations from total sulfur measurements.

The study will be of broad interest to the scientific community, as it points out that organosulfates have a much larger prevalence in atmospheric aerosols than generally thought. I expect that the study will lead to many new investigations on formation and occurrence of organosulfates in the atmosphere.

Thank you very much for this positive feedback.

In general, the study is well presented. However, the manuscript needs to be checked by a professional who is a native English speaker. One example of issues that needs to be corrected, is the wrong use of the word "exemplary" (e.g., Fig. 3 figure text). As there are several more instances of not grammatically correct English, the manuscript needs to be checked and corrected.

Initiated by this comment, we have completed a thorough language polishing by a professional service and have made the necessary corrections. Specifically, we have replaced the word 'exemplary' with 'representative' in the relevant sentences, including the figure caption of Fig. 3.

Line 15: ambient samples -> ambient aerosol samples

We have revised 'ambient samples' to 'ambient aerosol samples' in line 16 (the track-changes version) as recommended.

L17: I suggest to add ", respectively" after organosulfates

We have added ', respectively' after 'organosulfates' in line 19 (the track-changes version) as recommended."

L18-19: These sentences may not be clear to a more general reader.

We changed the sentence in line 19-21 (the track-changes version) as follows:

"We found a significantly larger organosulfate fraction appearing as a broad chromatographic peak in the charged aerosol detector. Confirming its origin from chromatographically non-resolved organosulfates, an all-ion-fragmentation experiment revealed specific sulfate-related ions."

We agree that readers who have a more general background need to read through the manuscript to understand the idea behind the "all-ion-fragmentation experiment" – but the word limit of the abstract will not allow for a more detailed explanation of AIF experiments.

L25: Up to 90% OA in atmospheric aerosols seems very high. Maybe it is more relevant to state a more general value.

We agree that 90% OA in atmospheric aerosols may seem high. We have revised this to reflect a more general and widely accepted value and added additional references to support the statement.

The new sentence was list in line 27-29 (the track-changes version) as follows:

"Organic aerosols (OA) are a major component in ambient PM, typically contributing 30-50% of the fine aerosol mass in the lower troposphere, though this can reach up to 90% in pristine environments⁴⁻⁷. ~~representing up to 90% of the fine aerosol mass in the lower troposphere~~⁴⁻⁷."

The new references are:

6. Zhang, Y., Cai, J., Wang, S., He, K. & Zheng, M. Review of receptor-based source apportionment research of fine particulate matter and its challenges in China. Sci. Total Environ. 586, 917–929 (2017).

7. Kanakidou, M. et al. Organic aerosol and global climate modelling: A review. Atmos. Chem. Phys. 5, 1053–1123 (2005).

L37: I was a bit surprised to see the paper by Le Breton et al. highlighted as the first paper presented in detail here, given that there are a number of earlier studies using UHPLC-MS as in the current work.

We appreciate your observation regarding the choice of highlighted studies. To improve the logical flow of the text, in the revised version we first cite a paper by Brüggemann et al., which discusses OS-related sample artifacts. We then reference the work of LeBreton et al., whose near-real-time method minimizes the formation of OS artifacts, demonstrating that the observed OSs are not purely artifacts. Finally, we cite the study by Ma et al., who report similar results to LeBreton et al., but with filter collection and UHPLC-HRMS measurements, further supporting the validity of our findings.

We changed the sentence in line 40-47 (the track-changes version) as follows:

"Several studies have analysed OSs in different environments, e.g. in the urban OA of Chinese megacities, in which rapid formation of OS during pollution events has been observed²⁸⁻³¹. However,

Brüggemann et al. reported significant sampling artifacts in the detection and quantification of monoterpene-derived OSs under atmospheric conditions³². LeBreton et al. used a Filter Inlet for Gases and Aerosols coupled to a Chemical Ionization Mass Spectrometer to measure OSs in near-real time, and therefore minimizing sampling artifacts, at a semi-rural site in Beijing. They quantified 17 single OSs, estimating their overall contribution to OA at 2%, suggesting that OSs are not merely artifacts of the sampling process³³. Similarly, a study conducted in Beijing using high-performance liquid chromatography (HPLC) combined with high-resolution mass spectrometry (HRMS) reported a ~4% contribution of OSs to OA²⁸.

The new reference is:

28. Ma, J. et al. Nontarget Screening Exhibits a Seasonal Cycle of PM2.5 Organic Aerosol Composition in Beijing. Environ. Sci. Technol. 56, 7017–7028 (2022).

32. Brüggemann, M. et al. Overestimation of Monoterpene Organosulfate Abundance in Aerosol Particles by Sampling in the Presence of SO₂. Environ. Sci. Technol. Lett. 8, 206–211 (2020).

L46-47: The use of camphor-10-sulfonic acid seems outdated, especially as the functional group is not a sulfate.

Thank you for highlighting the importance of using correct surrogates for OS quantification. It is indeed true that ionization efficiencies can vary significantly between different OSs under electrospray ionization. While compounds with the same functional group tend to exhibit more similar ionization efficiencies, the chemical structure of each compound also plays a crucial role. In this case, camphor-10-sulfonic acid remains a suitable surrogate standard when authentic OSs are unavailable.

Brüggemann et al. tested four authentic standards (2-hydroxy- α -pinene OS, 2-hydroxy-carene OS, limonaketone OS, and 2-hydroxy- β -caryophyllene OS) along with two surrogates (octyl sulfate and camphor-10-sulfonic acid), finding that calibration curves for these compounds could differ by up to two orders of magnitude (Fig. 1)¹. Despite containing a sulfonic group, camphor-10-sulfonic acid exhibited a medium response compared to the other tested compounds (Fig. 1)¹.

Fig. 1: Calibration curves of authentic terpene-derived OSs (solid lines) and commonly used surrogate standards (dashed lines). The inset shows the calibration curves on a linear scale. Abbreviations:

apinOS (2-hydroxy- α -pinene OS), carenOS (2-hydroxy-carene OS), limketOS (limonaketone OS), bcaryOS (2-hydroxy-b-caryophyllene OS), cmphrsulfon (camphor-10-sulfonic acid) and MSA (methylsulfonic acid). Figure from Brüggemann et al.¹

Additionally, we tested several authentic and surrogate OS standards under Orbitrap negative electrospray ionization mode using an isocratic gradient. The results showed that while camphor-10-sulfonic acid displayed a slightly lower response than the average of these compounds, it remained within 20% (Fig. 2). Therefore, in the absence of other authentic OS standards, we believe it is justified to use camphor-10-sulfonic acid as a surrogate for quantifying ambient OSs.

Fig. 2: Peak area (mean \pm SD) of ten OS standards (all at the same concentration levels of 2 ng/ μ L) measured with at an isocratic separation. The standards were prepared in one batch as an OS-mix. The mix contained: (1) Formaldehyde sodium bisulfite (CH₃SO₄Na), (2) Methyl sulfate sodium salt (CH₃SO₄Na), (3) Sodium ethyl sulfate (C₂H₅SO₄Na), (4) Butyl sulfate (C₄H₁₀SO₄Na), (5) Camphor-10-sulfonic acid (C₁₀H₁₆SO₄), (6) 2-hydroxy- α -pinene OS (C₁₀H₁₇SO₅), (7) Sodium octyl sulfate (C₈H₁₇SO₄Na), (8) Sodium n-decyl sulfate (C₁₀H₂₁SO₄Na), (9) Sodium dodecyl sulfate (C₁₂H₂₅SO₄Na), and (10) Tetradecyl sulfate sodium salt (C₁₄H₂₉SO₄Na). To test the stability of the instruments, we measured each standard concentration five times. The line (solid) is the average response of these compounds with a band (dash) of \pm 20%.

L76-77: *These first sentences are unclear. The general reader may not understand that the filter extraction procedure is similar, but sample preparation differs.*

To address this, we revised the sentence to explicitly state that while the filter extraction process is similar for all samples, the subsequent sample preparation is only for SPE.

We changed the sentence in line 90-93 (the track-changes version) as follows:

"In the field of atmospheric sciences, water or organic solvents are commonly used for liquid extraction of aerosol filter samples. The application of SPE for enrichment and fractionation of compounds is rarely practiced, but this sample preparation enables unique and novel analytical approaches. In the field of atmospheric sciences, where SPE techniques are not commonly applied,

~~we found unique analytical advantages over common filter extraction procedures. Here, we used...~~

L79-80: How does the sulfate group change acidity of nitrooxy-OSs?

The sulfate group significantly lowers the pKa of OSs and nitrooxy-OSs to approximately -2.4 to -4.6. Since the atmospherically relevant pH of aerosols typically ranges from 1 to 4, much higher than the pKa of OSs, all OSs and nitrooxy-OSs exist almost entirely in their dissociated (deprotonated) form under these conditions. As a result, the dissociation state of both OSs and nitrooxy-OSs is not affected by the specific pH within this range.

To improve clarity, we revised the sentence in lines 95–98 (tracked-changes version) as follows:

“A fundamental property of OSs, the terminal R-OSO₃H group, leads to a higher acidity (pKa ~ -2.4 to -4.6) for most of the (nitrooxy-)OSs compared to other acidic organic compounds in ambient air (e.g. nitro-phenols or organic acids), ensuring their almost complete dissociation under typical aerosol pH conditions (1 – 4)⁵¹.”

The new reference is:

51. Hyttinen, N., Elm, J., Malila, J., Calderón, S. M. & Prisle, N. L. Thermodynamic properties of isoprene-and monoterpene-derived organosulfates estimated with COSMOtherm. *Atmos. Chem. Phys.* 20, 5679–5696 (2020).

L83: Exemplary has a different meaning. Please correct. You probably mean that the sample is used as an example.

See above, we have replaced the word 'exemplary' with 'representative' in the relevant sentences, the new sentence in line 100-102 (the track-changes version) are as follows:

“In the following, we first describe an ~~representative~~representative ~~exemplary~~ sample from Handan with regard to our new enrichment and quantification approach, before discussing all analysed samples from both field sites.”

L86-87: This sentence is not clear.

We rewrote the sentence in line 104-106 (in the track-changes version) as follows:

“~~The two SPE fractions are diluted to the same concentration as the native extract. This enabled Dilution of the SPE fractions to the same concentration level as the native extract enables~~ a quantitative comparison and the evaluation of the absolute SPE-recovery of the single organic compounds detected by non-target analysis.”

L104: How large? Please be specific.

The molecular mass is up to ~350 Da, and the new sentence in line 124-126 (the track-changes version) are as follows:

“The recovered OSs span a large range regarding polarity and mass-to-charge ratio (m/z), ranging from relatively small (~150 Da) and polar (short RT) to large (~350 Da) and nonpolar compounds (long RT).”

L117: Can you state a standard deviation on the recovery?

Thank you for your suggestion, the overall recovery for the sample Handan 1, Handan 2, Handan 3, TO1, TO2, and TO3 are 0.71, 0.84, 0.67, 0.70, 0.78, and 0.80, respectively. Therefore, the standard deviation of the recovery is 6.6%, and we implement the SD in the manuscript. The new sentence in line 142-143 (the track-changes version) are as follows:

“With our SPE method, we obtained an overall mean OS (CHOS and CHNOS) recovery of 75% (with a standard deviation of 7%) for the six ambient aerosol samples (area-weighted mean, Fig. 2).”

L123: Often it is necessary to add the word “aerosol” when you talk about filter extracts to make the sentence more precise.

We implement the comment in the manuscript. The new sentence in line 148-150 (the track-changes version) is as follows:

“Overall, the SPE isolation method is capable of effectively isolating the OSs from complex aerosol filter extracts, thereby allowing the subsequent chemical analysis of the enriched fraction and the quantification of the OSs by the CAD.”

L140: What is the sensitivity of the CAD? Can this be described briefly, maybe in the experimental section?

CAD is sensitive in the sub-nanogram range, and we put this information in the method part. We revised the sentence in line 382-383 (the track-changes version) as follows:

“CAD has a sensitivity in the low-nanogram to high-microgram range (on column), which allows for the detection of trace levels of non-volatile compounds. Hence, we measured the concentrated second SPE elution with UHPLC-CAD/HRMS.”

L152: It seems strange to refer to Figure 3b before 3a.

Thank you for your comment. Upon review, we can confirm that Figure 3a is referred to before Figure 3b in the manuscript. Specifically, Figure 3a is first mentioned in **line 169**, while Figure 3b is referenced in **line 180** (as seen in the track-changes version). We believe the current structure is correct, but we appreciate your attention.

L145: Are other data available to support the interpretation? Can TO really be described as remote (which is typically further away from anthropogenic sources)?

We agree that the term “**remote**” is not entirely appropriate for describing the Taunus Observatory (TO). We changed the description to a “**rural**” site, given it located at 50.22° N, 8.44° E at 825 m altitude on top of Kleiner Feldberg in the Taunus Mountain range. In the North-West (the

dominating wind direction) the area is dominated by forested and agricultural areas. Due to rural setting, the observatory does have a European continental background character, allowing it to capture long-range transport of pollutants in addition to local emissions.

However, approx. 20 km south-east of the site is the city of Frankfurt (Main) in the center of the Rhine–Main area with several industrial sources including chemical industry. This site also enables assess the emissions from this densely populated region.

Therefore, while TO may not be truly remote, its location still allows for important observations of broader regional and continental air quality phenomena. We carefully review the entire manuscript, replacing "remote TO" and similar terms with "rural TO" and their equivalents.

L155: ambient filters -> suggest to change to: ambient aerosol filters - or similar.

We implement the comment in the manuscript. The new sentence in line 182-184 (the track-changes version) are as follows:

“Our regular liquid extraction of the ambient aerosol filters yielded no detectable peaks in the CAD, whereas strong signals appear in the OS fractions following SPE.”

L164: What is the M-effect?

In organic chemistry, the “–M-effect” refers to the mesomeric effect. It describes how electron-withdrawing or electron-donating groups attached to a conjugated system (like an aromatic ring) influence the distribution of electron density in the molecule through resonance.

The nitro group (–NO₂) of nitrosalicylic acid is an electron-withdrawing group, which exerts a negative mesomeric effect (–M-effect). This effect pulls electron density away from the aromatic ring through resonance.

This electron withdrawal increases the acidity of the carboxylic acid group on the same molecule because it stabilizes the negative charge on the conjugate base, making it easier for the carboxylic acid to lose a proton (H⁺).

To be clearer, we change the sentence in the line 192-196 (the track-changes version) as follows:

“From a chemical perspective, it is reasonable that these organic acids appear in the OS fraction, as the electron-withdrawing mesomeric effect (–M-effect) of the nitro group stabilizes the negative charge of the conjugated base of the carboxylic acid group (electron delocalization), thereby increasing the acidity of the nitrosalicylic acids. ~~as the M-effect of the nitro group on the aromatic ring increases the acidity of the carboxylic acid group of the nitrosalicylic acids.”~~

L190: Two days can hardly be called a period.

Thank you for your comment. Instead of referring to it as a 'period,' we changed it as a 'duration of two investigated winter days'. The new sentence in line 220-221 (the track-changes version) are as follows:

“The similar OS concentrations of the three samples indicate relatively constant conditions during the two investigated winter ~~period-days.~~”

L192: It is very unclear to the reader what this means. How were the sample selected?

We provided additional details on how the samples are selected. The revised text in line 222-225 (the track-changes version) are as follows:

“It is worth mentioning that the three TO aerosol filter samples used in this study were chosen from a large dataset of approximately 350 samples, collected from August 2021 to August 2022, due to their previously identified high occurrence of OSs. ~~the three TO filter samples were selected for this study from an extensive time series of measurements based on their previously identified high OS occurrence~~”

L194: A verb seems to be missing here. It is important to point out that these were three selected samples during winter, where previous studies have observed relatively high OS concentrations.

In the revised version, we have mentioned that “the three selected samples from winter and have observed relatively high OS concentrations” (see above), and we add the verb “a” in the text, The new sentence in line 225-227 (the track-changes version) reads as follows:

“The difference in concentrations between the two sites indicates, as expected, a much stronger anthropogenic influence on OS formation in the North China Plain, but still a significant abundance of OSs at a European rural station.”

L249- and Fig. 5: For the calculation of OM from organosulfates, was the sulfate group omitted?

No, we did not omit the sulfate group. we measure the organic carbon (OC) on all the aerosol filters. To estimate the organic matter (OM) content, we multiply the OC value for Handan aerosol filters by 1.6², and for TO aerosol filters, we multiply the OC value by 1.8³.

As for the OSs quantification, we calibrate the CAD by using eight different standards at five concentration levels. The 40 data points were fitted by using a second order polynomial function to get the calibration surface. Subsequently, we integrated the OS area detected in the ambient sample (which inherently includes the sulfate group) from the CAD chromatogram and input this value into the function to calculate its concentration. As a result, the OS fraction in OM represents an upper-limit estimation. The “other OM” (the legend of figure 5B) represent the OM value minus the sum of the resolved and unresolved OS.

L256: The North Sea is quite far away from TO compared to the closer densely populated and industrialized areas in Germany and Netherlands. The statement about shipping origin thus seems speculative.

Certainly, the North Sea is quite distant from the sampling site (TO). However, the molecular fingerprint indicates that most of the detected compounds belong to a homologous series of long-chain aliphatic OSs, such as C_nH_{2n+2}SO₄ and C_nH_{2n}SO₅ (Fig. 3) and these compounds appear actually at very low concentrations. Studies have suggested that ship emission are potential precursors of these types of OSs^{4,5}. At TO we also observe events of long-range transported pollution (unpublished work from our group) from South-Eastern Europe and Eastern Europe, hence, we believe that a significant source at the North Sea can be detected under the right transport

conditions at TO. Even the “ultra-low sulfur” heavy fuel oil, which is the class of oil that is allowed on the North Sea still contains up to 1000 ppm (0.1%) of sulfur. The back-trajectory analysis further supports the hypothesis of a North Sea origin (Fig. 4).

Fig. 3: Molecular fingerprints (m/z-RT plot, Van Krevelen diagram, and Kroll diagram) of the enriched OS fractions (TO 2: 14.01.2022, night-time).

Fig. 4: Backward trajectories for samples TO-2.

Certainly, there remain doubts about the exact origin of the OSs, therefore, we change the sentence in line 292-294 (the track-changes version) as follows:

“The homologue series of aliphatic OSs in this sample ~~indicates~~suggests an anthropogenic origin (Fig. S2e), ~~while~~ ~~Back~~trajectories indicate a possible marine origin from the North Sea (Fig. S16) with the aliphatic OS are likely being emitted by the shipping sector⁶⁰.”

L259: Again the reader is wondering how these three samples were selected.

See above (the answer for comments L192).

L261: This has previously been observed by e.g. Glasius et al., Composition and sources of carbonaceous aerosols in Northern Europe during winter, Atmospheric Environment, 173, 2018, pages 127-141.

Thank you for pointing out the study by Glasius et al. (2018), which we have now included as reference #61 in the line of 302 (the track-changes version).

61. Glasius, M. et al. Composition and sources of carbonaceous aerosols in Northern Europe during winter. Atmos. Environ. 173, 127–141 (2018).

L278: This seems like a very strong statement for apparently unpublished data.

We believe that OSs with an aliphatic carbon backbone are certainly of anthropogenic origin, while OS of biogenic origin have been described by many other studies in the past. It can be expected that a yearlong sampling that covers periods of high and low biogenic emissions (summer vs winter) can enable identifying the OSs from biogenic origin. Therefore, we don't think that this is such a strong statement, but rather a realistic outlook what could be achieved with our described method in future work.

Figure 1: I suggest to make larger dots in the legend, so they are easier to see.

We have updated Figure 1 accordingly (in line 131 of the track-changes version), with larger dots in the legend to enhance clarity. New figure sees below:

Figure 2: In the description of the aerosol filter sample extraction in the figure, I suggest to list the composition of the extraction solvent.

We have updated Figure 2 accordingly (in line 152 of the track-changes version), with the composition of the extraction solvent. New figure sees below:

Figure 2. Schematic of the extraction and enrichment procedure. Relative abundance (%) of summed peak areas ((-)ESI of the different compound classes in the native extraction (A), diluted OS fraction (B) and enriched OS fraction (C). The y-axis shows the relative abundance of the compound classes, normalised to the most intense sample (native extraction of the Handan 3 sample). For better clarity, the left y-axis refers to the Handan samples (black; 0-100%), and the right y-axis to the Taunus Observatory (TO) samples (pink; 0-20%). The samples used were Handan 1: 18.10.2018, daytime; Handan 2: 21.10.2018, daytime; Handan 3: 21.10.2018, night-time; TO 1: 11.12.2021, daytime; TO 2: 14.01.2022, night-time, and TO 3: 28.02.2022, night-time. In order to make the native extraction (A), the diluted OS fraction (B) and the enriched OS fraction (C) comparable, we accounted for the enrichment factor by the SPE (~240) and the different experimental setup in (C): (A) and (B) were acquired using HPLC-HRMS (100% of mobile phase flow into the MS) and (C) by using HPLC-HRMS/CAD (16.5% of mobile phase flow into the MS). The consistently larger relative abundance of OSs in fraction (C), therefore, solely originates from additionally detected OSs following SPE. Created in BioRender. Reininger, N. (2025) <https://BioRender.com/n70h222>.

Figure 3: Is CHOP a common group of compounds in aerosols?

Yes, CHOP compounds, which refer to organic compounds containing carbon (C), hydrogen (H), oxygen (O), and phosphorus (P), are common in aerosols, though they are less commonly discussed compared to other groups like OSs or organonitrates. Due to their large anthropogenic sources, being classified as High Production Volume (HPV) chemicals, they are widely used as flame retardants or plasticizers. Generally, the primary emitted CHOP compounds (Fig. 5) don't have acidic protons (e.g., TCEP, TCPP, TPhP, TBEP), however, their transformation products tend to have hydroxyl or carboxyl group (e.g., TCEP-1, TCEP-21, TCPP-38, TCPP-9, TPhP-6, TPhP-8)⁶, which enables the detection of these compounds in negative electrospray ionization mode (Fig. 6).

Fig. 5: some of the primary emitted CHOP compounds. Figure modified from Liu et al.⁶

Fig. 6: some of the transformation products (which contain acid protons) of primary CHOP compounds. Figure modified from Liu et al.⁶

Reviewer 2

Summary and Recommendation:

This manuscript examines the types and amounts of organosulfates (OSs) found in PM_{2.5} samples collected from Handan, China (urban) and Hesse, Germany (mountain top site that is remote) by using solid phase extraction (SPE) as a sample preparation step before RPLC-CAD-HRMS analysis. Although SPE has been rarely used in aerosol chemistry research, it helps in aiding the identification and quantification of OSs when using RPLC-CAD-HRMS. I should note that SPE was originally used in earlier work that analyzed PM_{2.5} samples collected from the SE USA (e.g., Gao et al., 2006, JGR-Atmospheres), but it was later neglected due to it removing small water-soluble OSs (such as the isoprene-derived OSs) in one of the washing steps. This latter recognition was so important to isoprene-derived SOA because using SPE caused these OSs to go initially unrecognized. With that said, to my knowledge, this study uses RPLC coupled with both CAD with ESI-HRMS detection. The CAD offers incredible potential to quantify OSs without the need for all of the authentic standards being available. In atmospheric chemistry, the lack of available authentic standards has been a huge hindrance, preventing models

from being updated with the latest multiphase chemical processes that lead to low-volatility OSs. Overall, I find this manuscript exciting and mostly well-written, especially for atmospheric and aerosol chemists. Some readers not well-versed in analytical chemistry might find this paper somewhat difficult to read. However, I hope the Editor won't find this a reason to reject the paper. I do think the demonstration of a large un-resolved OS peak in the chromatogram is notable and appears to contribute a large OA mass to PM_{2.5} collected during the winter seasons in China and Germany. I have several major comments below that should be carefully addressed by the authors before publication in Nature Communications can be fully considered. Due to the nature of my comments below, I recommend the manuscript be reconsidered after major revisions.

We appreciate the recognition of the novelty and importance of our study.

Major Comments:

1.) Filter collection artifacts:

There has been some important work to recently come out of Hartmut Hermann's lab (and I believe Marianne Glasius' lab) showing that filter sampling (especially with media like quartz fiber filters) is prone to positive sampling artifacts that lead to the production of certain OSs like m/z 294 (one of the nitrooxy OSs). Since high-volume filter samplers cannot use denuders (due to the high flowrates used) to remove gas-phase SO₂ and NO_x, it is possible that NO_x and SO₂ can absorb from the gas phase and react on these filter types yielding unwanted reactions that lead to OSs. Knowledgeable readers might wonder how much of the mass you quantify is related to these positive sampling artifacts?

The work by Brüggemann et al. (from Prof. Hartmut Hermann's group) indeed highlights the formation of organosulfates (OSs) on quartz fiber filters during chamber experiments in the presence of SO₂.⁷ However, the conditions in their study differ significantly from those at our rural background site, where SO₂ concentrations are generally very low (e.g. the SO₂ concentration of TO during our sampling day was lower than the detection limit of the SO₂ detector). We infer that most OSs observed at TO station form during atmospheric transport. For example, backward trajectory analysis for the TO₂ (Fig.7a also in supplementary Information Fig. S16a) and TO₃ (Fig. 7b also in supplementary Information Fig. S16b) samples indicates distinct air mass origins. For TO₂, the air masses originated from the North Sea, suggesting a marine influence, with aliphatic OSs potentially emitted by the shipping sector (Fig. 3). In contrast, TO₃ air masses originated in eastern Germany, southern Poland, and northern Czech Republic—key regions in Europe associated with coal combustion by power plants. Particularly, the approximately one-day atmospheric transport time from the source region is twice the duration of the filter sampling period, suggesting minimal OS artifact formation in the TO filter samples.

Fig. 7: Backward trajectories for samples TO-2 (a) and TO-3 (b). Backward trajectories for Taunus Observatory (825 m.a.s.l.) starting every hour for the 12-hour filter sampling duration. Duration of backward trajectories is 48 hours and was calculated based on the online web version of the NOAA Hysplit Model.⁸

We also extracted the nitrooxy-OSs ion (m/z 294) from the LC-MS chromatograms of the solvent blank, ambient quartz filter blank, and ambient quartz filter sample (Fig. 8). The signal for all nitrooxy-OSs isomers is significantly higher in the ambient sample than in the solvent and filter blanks. Hence, this indicates the presence of OSs in the aerosol rather than their formation as filter artifacts.

Fig. 8: Comparison of extracted ion chromatograms (EICs) of nitrooxy-OSs (m/z 294.0653) in the solvent blank, ambient quartz filter blank, and ambient quartz filter sample (Handan, October 21, 2018, nighttime), measured by HPLC-HRMS.

We also implemented several measures to minimize their impact. For each site, we collected multiple filter blanks to subtract the background signal. These blanks were exposed for the same

duration as the real filters, with the only difference being that the real filters operated with the pump running for 11.5 hours. Additionally, during data processing with Compound Discoverer, we applied a stringent threshold to ensure data reliability, considering only signals in the ambient filters that were at least ten times higher than those in the blanks as real signals.

Overall, we believe that positive sampling artifacts have minimal influence on our quantified mass. However, we cannot entirely rule out the possibility of sampling artifacts, therefore, we change the sentence in the line 297-303 (the track-changes version) as follows:

“Furthermore, we cannot rule out that a fraction of OSs is formed during filter sampling, although the SO₂ concentrations at this rural background station are usually below quantifications limits for standard detectors. Hence, we infer that the majority of OSs observed at rural and remote stations actually forms during atmospheric transport. Analysis of the backward trajectories for TO 3 indicate that the air masses originated from eastern Germany, southern Poland and the northern Czech Republic, which are Europe’s hotspots of coal combustion by power plants (Fig. S16)⁶¹. The time of atmospheric transport from this source area of around one day is twice as long as the actual filter sampling period, which speaks against significant OS artifact formation for the TO filter samples.”

Also see answer to the first reviewer L37

To improve the logical flow of the introduction, we cite some new references in line 40-47 (the track-changes version):

We first cite a paper by Brüggemann et al., which discusses OS-related sample artifacts. We then reference the work of LeBreton et al., whose near-real-time method minimizes the formation of OS artifacts, demonstrating that the observed OSs are not purely artifacts. Finally, we cite the study by Ma et al., who report similar results to LeBreton et al., but with filter collection and UHPLC-HRMS measurements, further supporting the validity of our findings.

We changed the sentence in line 40-47 (the track-changes version) as follows:

“Several studies have analysed OSs in different environments, e.g. in the urban OA of Chinese megacities, in which rapid formation of OS during pollution events has been observed^{28–31}. However, Brüggemann et al. reported significant sampling artifacts in the detection and quantification of monoterpene-derived OSs under atmospheric conditions³². LeBreton et al. used a Filter Inlet for Gases and Aerosols coupled to a Chemical Ionization Mass Spectrometer to measure OSs in near-real time, and therefore minimizing sampling artifacts, at a semi-rural site in Beijing. They quantified 17 single OSs, estimating their overall contribution to OA at 2%, suggesting that OSs are not merely artifacts of the sampling process³³. Similarly, a study conducted in Beijing using high-performance liquid chromatography (HPLC) combined with high-resolution mass spectrometry (HRMS) reported a ~4% contribution of OSs to OA²⁸.

The new references are:

28. Ma, J. et al. Nontarget Screening Exhibits a Seasonal Cycle of PM_{2.5}Organic Aerosol Composition in Beijing. Environ. Sci. Technol. 56, 7017–7028 (2022).

32. Brüggemann, M. et al. Overestimation of Monoterpene Organosulfate Abundance in Aerosol Particles by Sampling in the Presence of SO₂. Environ. Sci. Technol. Lett. 8, 206–211 (2020).

2.) Lines 29-31: The authors should cite the earliest papers here for OSs being discovered as an important subclass of OA, including the following:

Romero and Oehme (2005, *J. of Atmos. Chem.*) - DOI: 10.1007/s10874-005-0594-y

Surratt et al. (2007, *ES&T*) - DOI: 10.1021/es062081q

linuma et al. (2007, *Atmos. Environ.*) - <https://doi.org/10.1016/j.atmosenv.2007.03.007>

linuma et al. (2007, *ES&T*) - <https://doi.org/10.1021/es070938t>

Surratt et al. (2008, *JPCA*) - <https://doi.org/10.1021/jp802310p>

The Surratt et al. (2008, *JPCA*) study estimated that upwards of 30% of particulate OM in Hungary could be OSs.

Additional work by Betsy Stone's group has also estimated OS contributions to OM, such as <https://doi.org/10.5194/acp-19-3191-2019>. In that study, they estimated 16.5% of PM_{2.5} OC as OSs.

Thank you for pointing out these studies, we have included these studies as references #8-13 in the line of 32 (the track-changes version). The revised sentence are as follows:

“Organosulfates (OSs) are an important subclass of OA⁸⁻¹³, comprising up to 30% of the OA¹². They...”

The new references are:

8. Romero, F. & Oehme, M. Organosulfates - A new component of humic-like substances in atmospheric aerosols? *J. Atmos. Chem.* 52, 283–294 (2005).

9. Surratt, J. D. et al. Evidence for organosulfates in secondary organic aerosol. *Environ. Sci. Technol.* 41, 517–527 (2007).

10. linuma, Y. et al. Evidence for the existence of organosulfates from β -pinene ozonolysis in ambient secondary organic aerosol. 41, 6678–6683 (2007).

11. linuma, Y., Müller, C., Böge, O., Gnauk, T. & Herrmann, H. The formation of organic sulfate esters in the limonene ozonolysis secondary organic aerosol (SOA) under acidic conditions. *Atmos. Environ.* 41, 5571–5583 (2007).

12. Surratt, J. D. et al. Organosulfate formation in biogenic secondary organic aerosol. *J. Phys. Chem. A* 112, 8345–8378 (2008).

13. Hettiyadura, A. P. S., Al-Naiema, I. M., Hughes, D. D., Fang, T. & Stone, E. A. Organosulfates in Atlanta, Georgia: Anthropogenic influences on biogenic secondary organic aerosol formation. *Atmos. Chem. Phys.* 19, 3191–3206 (2019).

3.) Lines 31-32: I'm a bit surprised the authors didn't cite any work on the reactive uptake of isoprene epoxydiols (IEPOX) yielding OSs in PM_{2.5}. This is one of the major semi-volatile oxidation products out there that has helped clue us in as a research community on the role of reactive uptake (multiphase chemistry) onto wet acidic sulfate aerosols that yield OSs.

Thank you for highlighting the relevance of the reactive uptake of isoprene epoxydiols (IEPOX) as a key pathway for OS formation from isoprene. This mechanism indeed plays a crucial role in advancing our understanding of multiphase chemistry on acidic sulfate aerosols, and being especially relevant for high-isoprene emission regions such as the Amazon or the south-east US.

We have now included relevant references of the studies discussing the IEPOX yielding OSs. The sentence in the line of 34-36 (the track-changes version) now reads as follows:

“The reactive uptake of semi-volatile organic compounds, particularly isoprene epoxydiols (IEPOX), onto acidic particles can result in the formation of low-volatile OSs and, consequently, add to the OA mass^{9.16.21-2346.24}.”

The newly cited references:

9. Surratt, J. D. et al. Evidence for organosulfates in secondary organic aerosol. *Environ. Sci. Technol.* 41, 517–527 (2007).

22. Riva, M. et al. Chemical Characterization of Secondary Organic Aerosol from Oxidation of Isoprene Hydroxyhydroperoxides. *Environ. Sci. Technol.* 50, 9889–9899 (2016).

23. Surratt, J. D. et al. Chemical composition of secondary organic aerosol formed from the photooxidation of isoprene. *J. Phys. Chem. A* 110, 9665–9690 (2006).

4.) *Lines 32-33: Similar to earlier lines in the introduction, simply citing review articles for statements like this doesn't seem appropriate. I think the authors should be citing some of the original studies that demonstrated the role of SO₂ emissions in enhancing SOA from BVOC oxidations.*

We agree that citing original studies demonstrating the role of SO₂ emissions in enhancing SOA formation from BVOC oxidations will strengthen the manuscript.

In response, we have included, in addition to the important review article by Brüggemann et al., a few pivotal references in line of 37 (the track-changes version). Certainly, there are more publications on this topic, but we also have to consider that the journal has a certain limit on the number of references.

The newly cited references:

12. Surratt, J. D. et al. Organosulfate formation in biogenic secondary organic aerosol. *J. Phys. Chem. A* 112, 8345–8378 (2008).

25. Surratt, J. D. et al. Reactive intermediates revealed in secondary organic aerosol formation from isoprene. *Proc. Natl. Acad. Sci. U. S. A.* 107, 6640–6645 (2010).

5.) *Lines 36-49: This paragraph is very European- and Asian-centric. I think this paragraph is fine, but a lot of prior OS work has also been done in the USA trying to quantify OSs in PM_{2.5} there. There has been a lot of work by the Surratt Group (UNC) and Stone Group (University of Iowa) on this topic using LC/MS methods and synthesizing authentic compounds. Furthermore, Jian Yu's Group (HKUST) in Hong Kong has also estimated OS contributions in PM_{2.5} and has synthesized authentic compounds as well.*

To address this point, we have revised the paragraph in the line 48-52 (the track-changes version) to ensure a more comprehensive and balanced representation of global research efforts in this field. Similar to the previous answer, there are more publications from America, but we need to consider that the journal imposes a limit on the number of references.

The new text now as follows:

“In urban and rural OA, studies have shown significant seasonality in both biogenic and anthropogenic OSs at various sites, with biogenic OSs peaking in summer and anthropogenic OSs in winter^{34,35}. LeBreton et al. quantified 17 single OSs and estimated their overall contribution as being 2% to the OA at a semi-rural site in Beijing by using the Filter Inlet for Gases and Aerosol (FIGAERO) coupled to a chemical ionisation mass spectrometer²⁴. In marine OA, in the Yellow and Bohai Seas, the contribution of biogenic OSs ranges from 0.04 to 6.9%³⁶.”

The newly cited references:

34. Meade, L. E. et al. Seasonal variations of fine particulate organosulfates derived from biogenic and anthropogenic hydrocarbons in the mid-Atlantic United States. *Atmos. Environ.* 145, 405–414 (2016).

35. Chen, Y. et al. Seasonal Contribution of Isoprene-Derived Organosulfates to Total Water-Soluble Fine Particulate Organic Sulfur in the United States. *ACS Earth Sp. Chem.* 5, 2419–2432 (2021).

6.) *Lines 92-93: How comfortable are the authors assuming that the majority of OSs are stable in the presence of a strong acid? Work from Matt Elrod's group (Oberlin College) has suggested that the stability of OSs depends on whether the OS is primary or tertiary. Did the authors consider testing their approach with different OS standards to confirm the stability of OS compounds?*

To confirm the stability of OSs under the acidic conditions for SPE elution used in our study, we conducted experiments with eight OS standards (one secondary OS (5) and one tertiary sulfonate (4)) with varying molecular structures. Figure 9 presents the extracted-ion chromatograms (EICs) for individual standards before and after SPE, alongside their molecular structures and recoveries.

We observed that small, polar compounds (e.g., C1 and C2 compounds) either had lower recoveries or reacted under acidic conditions, consistent with observations noted in the manuscript (marked version, lines 126–128). In contrast, the remaining compounds demonstrated high stability, with recoveries close to 100% using our SPE method involving 0.5% HCl (Fig. 9). Additionally, no differences in recovery were observed between primary and secondary OS (tertiary sulfonate). The retention times for all standards remained within a 0.05-minute tolerance, indicating no chemical transformation of these compounds.

Fig. 9: Comparison of extracted ion chromatograms (EICs) and molecular structures of the eight OS test standards, measured by HPLC-HRMS, before and after SPE isolation (the post-SPE elution sample was diluted by the theoretical SPE enrichment factor to enable the direct comparison with the pre-SPE sample).

Furthermore, the nontarget analysis of ambient representative filter (Handan 1) shows 72% of the compounds in the OS fraction had an absolute recovery greater than 50% compared to the native extraction. Some OSs exhibited poorer recoveries, ranging from 10% to 50%, with relatively low signal intensities (Fig. 10a, in Supplementary Information Fig. S1a). By extracting this sample via SPE in triplicate, we determined that the intensity-weighted average relative standard deviation (RSD) of the SPE recovery for OSs was approximately 15%, with larger uncertainties for low-intensity compounds (Fig. 10b, in Supplementary Information Fig. S1b).

Fig. 10: Evaluation of the SPE isolation recovery (a) and reproducibility (b) for individual compounds using a PM2.5 filter extract from Handan (18.10.2018, daytime).

These results clearly demonstrate that the majority of OSs analyzed in this study remained stable under the acidic SPE conditions employed. The high recoveries and consistent retention times of the tested standards, combined with the highly reproducible nontarget screening results of

ambient OSs, show that our approach effectively captures most OS compounds with minimal degradation or transformation.

To make the sentence more precise, we revised the sentence in lines 111–113 (tracked-changes version) as follows:

“Subsequently, we ~~The majority of OSs are chemically stable which, thus, allowed the~~ used of a ~~strong-diluted hydrochloric~~ acid dissolved in methanol to elute the ~~OSs~~ ~~m-OSs~~ from the cartridge. ~~We did not observe any evidence for degradation of the OSs, confirming their chemical stability at low pH⁵².”~~”

7.) Figure 1d:

As shown in Figure 1d, there are still some OS compounds that elute through what they call "flow-through." Are these accounted for? One thing I want to note to the authors is that in Gao, Surratt, et al. (2006, JGR-Atmospheres), they employed an SPE separation to measure SOA. When they did this OSs were missed, especially the very water-soluble types (like the isoprene-derived OSs). Later Surratt et al. (2007, ES&T) found that by not using SPE, but rather a polar-end capped C18 column, they could measure the isoprene-derived OSs along with the larger monoterpene-derived OSs. Later, Betsy Stone's group (University of Iowa) and later Surratt's group (UNC) showed that HILIC separation is much more effective in resolving and quantifying isoprene-derived OSs and other water-soluble and smaller OSs. I'm curious, do the authors worry they are still missing water-soluble OSs that are not well retained on C18 columns? I noted the authors conducted reverse-phase liquid chromatography (RPLC) to separate and measure particulate OSs. However, it might be better for all of us in the community to use both HILIC and RPLC separations as orthogonal methods to better separate, characterize and quantify the OSs that range from water-soluble to less water-soluble. Do you agree? It might be worth mentioning the limitations of RPLC in past work and why HILIC methods have come about for the smaller OSs.

We appreciate that the reviewer mentions prior work on OS separation methods, particularly highlighting the challenges of analyzing highly water-soluble OSs with RPLC and the advantages of alternative methods like HILIC. We agree that HILIC is the most favorable method for the separation of polar organic compounds.

Accounting for flow-through OS compounds in Figure 1d:

The "flow-through" fraction observed in Figure 1d was not included in the recovery calculation. Since the solvent for the flow-through is 2% MeOH in H₂O, it is not possible to concentrate further, and it cannot reach the detection limit of the CAD. Additionally, this fraction only contains a very small portion of the CHOS compounds. We have also highlighted this limitation of the method in the manuscript (the track-changes version, lines 126–129).

OS Retention on SPE MAX cartridge:

In this study, however, we use a mixed-mode anion-exchange (MAX) cartridge combined with a reversed-phase sorbent (Fig. 11). The quaternary ammonium cation (functional group ③) in the cartridge interacts with the highly acidic terminal R-OSO₃H group of OSs, ensuring strong retention of OSs.

Fig. 11: the polymer skeleton of the Mixed-mode Anion-exchange cartridge with a reversed-phase sorbent.

To evaluate the SPE recovery of OSs from isoprene, we performed a new dedicated experiment. We used an oxidation flow reactor to generate isoprene-derived OSs. Using a PAM-OFR, isoprene was oxidized by O_3 (2 ppm) and photooxidation (OH was generated by UV light ($\lambda = 254$ nm) inside the OFR). Additionally, we inject SO_2 (55 ppb) at relative humidity $\sim 60\%$. The SOA exiting the OFR passed through two 50 cm denuders filled with charcoal to remove reactive gas-phase compounds. SOA particles were collected on glass fiber filters at a flow rate of 3 L min^{-1} over 120 minutes.

We extracted the filters and applied the SPE method described in the paper, using this time a HILIC column for the separation of isoprene OSs. A comparison of compound intensities across different fractions, based on non-target analysis, revealed that the second SPE elution recovered only $\sim 30\%$ of the OSs present in the native elution. However, the missing OSs were not detected in the flow-through fraction (Fig. 12). We infer that these OSs were either strongly retained on the anion-exchange cartridge or they are more reactive compared to longer chain OS.

Fig. 12: Summed peak areas of different compound classes in the isoprene-derived OS sample across the native extraction, first elution, second elution, and flow-through, analyzed using the HILIC column (both first and second elutions were diluted by the theoretical SPE enrichment factor to enable the direct comparison of areas).

We are currently working to isolate isoprene-derived OSs by testing different elution strategies for the SPE, in case the isoprene-OS are retained very strongly to the solid phase material. However, such detailed method development is beyond the scope of this manuscript. Instead, we focus on the scientific novelty of our findings, specifically the discovery of a significant fraction of chromatographically unresolved OSs in ambient aerosols, which appear throughout the chromatogram of the RPLC-column.

Discussion of the limitations of RPLC:

For this study, we focused on RPLC because it allowed for effective separation and quantification of the majority of OSs detected in ambient aerosol samples. However, we agree that using HILIC as an orthogonal method alongside RPLC would provide a more comprehensive characterization of OSs. Therefore, revisions were made to the manuscript in the line of 126-129 (the track-changes version):

“However, we did note that the used SPE cartridge provides a low recovery of weakly retained some small and highly polar OSs (e.g. C₂-C₃), which may include certain isoprene-derived OSs, an important class of atmospheric OSs. To a lesser extent, reversed-phase LC (used in this study) is also less effective for separating these compounds, highlighting the complementary role of hydrophilic interaction LC methods in analyzing smaller OSs⁵⁴.”

We have added a new reference #54 in the line 129 (the track-changes version) to support our discussion:

54. Cui, T. et al. Development of a hydrophilic interaction liquid chromatography (HILIC) method for the chemical characterization of water-soluble isoprene epoxydiol (IEPOX)-derived secondary organic aerosol. *Environ. Sci. Process. Impacts* 20, 1524–1536 (2018).

We added a sentence to the manuscript in the line of 317-318 (the track-changes version) and included the Fig.12 in the supplementary Information as Figure S17:

“Especially, isoprene-derived OSs might still be underestimated using the described method, as we found the SPE-recovery for these polar compounds is around 30% (Fig. S17).”

8.) *Line 117-118: How do the authors know they extracted all of the OS compounds and mass from their filters using 98% water and 2% methanol by volume solvent only? Were standards of OSs also examined through their extraction procedures? I think this would be helpful to include. Could larger OSs be missed if not soluble in water? Also, with water, do you have to worry about the hydrolysis of OSs?*

We agree it is important to evaluate extraction efficiencies of the used solvent in this study. Since OSs are highly water-soluble, we are confident that a water-based extraction solvent is effective for extracting the majority of OSs. Of course, the extraction efficiencies may be lower for non-polar OSs, which is why we also tested the method with additional experiment by using authentic standards, as described below:

We prepared a solution of eight OS standards with varying molecular structures (see Fig. 9 question 6) and spiked the same amount of the standard mix onto the following filters: 1) Handan ambient blank (quartz fiber), 2) Handan ambient sample (quartz fiber), 3) TO ambient blank (glass fiber), and 4) TO ambient sample (glass fiber). These filters were then extracted twice with a water-based solvent (2% MeOH in H₂O), and the extracts were combined. The extracts were analyzed by LC-HRMS together with the pure standard solution to evaluate the extraction efficiency for different filter materials. The extracted ion chromatogram of the eight standards from the five

measurements (LC-HRMS chromatogram) are displayed in Fig. 13. We calculated the extraction efficiency of the standard mix for the four filter extracts based on the pure standards measurement (Fig. 14).

Using our extraction procedure, we found that the recovery of the standards spiked onto different filter materials was consistently above 80%, except for dodecyl sulfate ($C_{12}H_{25}SO_4^-$) on the Handan ambient filter, which was slightly below 80% (Fig. 14).

Additionally, we considered the possibility of hydrolysis during extraction but observed no significant degradation of OSs under the conditions used, as also verified through the stability of standards (Fig. 9). These results support the reliability of our extraction method in capturing the majority of OS compounds without significant loss or reaction.

Fig. 13: Comparison of extracted ion chromatograms for the eight OS standards across different filter materials.

Fig. 14 Comparison of the responses of the eight OS standards across different filter materials. The black line is normalized to the pure standard measurement.

9.) Lines 121-122: I agree that the recoveries can be affected by what the authors state here. But what about also the matrix of the sample? Could that affect the recovery from the filters? For example, it is known that particulate PAHs can be hard to extract from certain filters depending on the matrix of the filter and the other components collected on the filter.

We also tested our extraction procedure across different filters and filter materials and found that the recovery of almost all the standards spiked onto these filters was consistently above 80%.

See the answer to question 8.

10.) Lines 144-145: The authors note that there are OSs that are aliphatic and of anthropogenic origin. Are the authors aware of work by Riva et al. (2016, ACP)? This study demonstrated OSs form from the oxidation of alkanes in the presence of wet acidic sulfate aerosols. Riva et al. (2015, ES&T) also demonstrated OSs form from PAH oxidations in the presence of sulfate aerosols. Notably, in this latter study, they also observed sulfonates. Did you by chance observe any sulfonates in your PM_{2.5}, possibly from the wintertime samples? What about other oxidation states of sulfur?

Thank you for highlighting the studies by Riva et al. (2016, ACP) and Riva et al. (2015, ES&T). We are aware of these publications, which strongly support our hypothesis that some OSs are aliphatic and of anthropogenic origin. Accordingly, we have cited these studies in the revised manuscript (the track-changes version, line 173). The new references #22 and #55:

22. Riva, M. et al. Chemical Characterization of Secondary Organic Aerosol from Oxidation of Isoprene Hydroxyhydroperoxides. *Environ. Sci. Technol.* 50, 9889–9899 (2016).

55. Riva, M. et al. Chemical characterization of organosulfates in secondary organic aerosol derived from the photooxidation of alkanes. *Atmos. Chem. Phys.* 16, 11001–11018 (2016).

We observed some sulfonates in our winter samples, such as $C_5H_{12}SO_3$ and $C_8H_{18}SO_3$ (Fig. 15, daytime sample from Handan on 18-10-2018). However, their intensities were lower compared to those of OSs. All other sulfur-containing compounds identified in our samples were OSs, with sulfur in an oxidation state of +6.

Fig. 15: Extracted ion chromatograms for $C_5H_{12}SO_3$ and $C_8H_{18}SO_3$.

11.) Minus the comment I made about the filter sampling artifacts above, I wonder if this method would show even higher OS fractions in fine particulate OM during the summer seasons when BVOC-derived oxidation products are known to yield substantial amounts of OS in the presence of sulfate? The fact the authors show fairly high amounts in winter, tells me the mass fractions could even be higher than previously thought in summer....

To explore this hypothesis, we conducted a new experiment involving solid-phase extraction (SPE) and HPLC-CAD-HRMS analysis using a summer season sample collected on July 2, 2019, from Handan, China. Our results revealed a larger fraction of unresolved OS in this sample (Fig. 16). This finding supports the idea that BVOC-derived oxidation products significantly contribute to OS formation in the presence of sulfate under summer (haze) conditions.

Unfortunately, our group currently lacks the instrumentation required to measure organic carbon (OC) on the filters, preventing us from calculating the OS fractions in organic matter. As part of our future work, we plan to analyze yearlong filter samples with this method. This will help distinguish purely anthropogenic from mixed biogenic/anthropogenic OS contributions, providing a more comprehensive understanding of seasonal variations across different locations.

Fig.16: Comparison of the chromatograms of the enriched OS fraction of the Handan filter (July 2, 2019, black) and blank (red).

12.) Some readers or reviewers might ask why the authors didn't try this RPLC-DAD-HRMS method on a well-characterized lab-generated SOA type like IEPOX SOA before analyzing field samples? The reason I raise this point is for IEPOX SOA our community has an analytical method (i.e., HILIC/ESI-HR-QTOFMS from both the Stone and Surratt Groups) and authentic standards (from the Surratt Group) to compare your quantitative method with. Furthermore, lab studies typically can denude unwanted gases (like SO₂/NO_x) before collecting onto Teflon filters (which are less prone to positive sampling artifacts like quartz).

See answer 7. We generated an isoprene SOA filter, but encountered difficulties eluting the OSs from the MAX SPE cartridge. We are currently optimizing the method, such as adjusting the acid used to elute the cartridge. However, we believe that a detailed method development will be better suited for a separate publication.

That said, we recognize the potential of HILIC for more effectively separating isoprene-derived OSs compared to RPLC (Fig. 17), as HILIC offers a complementary approach for polar compound analysis. We are actively working on integrating our SPE method with HILIC-CAD-HRMS to enhance the separation and detection of isoprene-derived OSs.

Fig. 17 Comparison of the effects of separation of C₅H₁₂SO₇ using the RPLC and HILIC.

References:

1. Brüggemann, M. *et al.* Quantification of known and unknown terpenoid organosulfates in PM10 using untargeted LC–HRMS/MS: contrasting summertime rural Germany and the North China Plain. *Environ. Chem.* **16**, 333–346 (2019).
2. Wang, Y. *et al.* Comparative Study of Particulate Organosulfates in Contrasting Atmospheric Environments: Field Evidence for the Significant Influence of Anthropogenic Sulfate and NO_x. *Environ. Sci. Technol. Lett.* **7**, 787–794 (2020).
3. D. Surratt, J. *et al.* Organosulfate formation in biogenic secondary organic aerosol. *J. Phys. Chem. A* **112**, 8345–8378 (2008).
4. Qi, L. *et al.* Molecular characterization of atmospheric particulate organosulfates in a port environment using ultrahigh resolution mass spectrometry: Identification of traffic emissions. *J. Hazard. Mater.* **419**, 126431 (2021).
5. Huang, L. *et al.* Biogenic and Anthropogenic Contributions to Atmospheric Organosulfates in a Typical Megacity in Eastern China. *J. Geophys. Res. Atmos.* **128**, e2023JD038848 (2023).
6. Liu, Q. *et al.* Uncovering global-scale risks from commercial chemicals in air. *Nat.* **2021** 6007889 **600**, 456–461 (2021).
7. Brüggemann, M. *et al.* Overestimation of Monoterpene Organosulfate Abundance in Aerosol Particles by Sampling in the Presence of SO₂. *Environ. Sci. Technol. Lett.* **8**, 206–211 (2020).
8. Stein, A. F. *et al.* NOAA’s HYSPLIT Atmospheric Transport and Dispersion Modeling System. *Bull. Am. Meteorol. Soc.* **96**, 2059–2077 (2015).

Manuscript ID: NCOMMS-24-43312B

Manuscript Type: Article

Collection: Mass spectrometry method development

Response to reviewer

Reviewers' Comments to the Authors:

Reviewer 1

The authors have done a very good job addressing the comments from the reviewers. However, I have a few additional comments to the response.

Thank you for your positive assessment and for taking the time to review our responses.

Line 18-19

"We found a significantly larger organosulfate fraction appearing as a broad chromatographic peak in the charged aerosol detector." You do not find peaks inside the charged aerosol detector, but maybe in the chromatograms from it. Please use correct wording.

We changed the sentence in line 19-20 (the track-changes version) as follows:

"We found a significantly larger organosulfate fraction appearing as a broad chromatographic peak in the chromatograms from the charged aerosol detector."

Regarding the investigation of responses of different OS standards, it can be discussed whether the first compound formaldehyde sodium bisulfite has a sufficient response or it should be excluded from the calculation of average response and variation, when evaluating the suitability of camphor-sulphonic acid as reference standard.

Thank you for pointing out the low response of formaldehyde sodium bisulfite. However, we believe its lower response in ESI is real, and we cannot rule out that other organosulfates exhibit deviating response from the average. Hence, including this compound accounts for the overall larger uncertainty with ESI quantification.

Compared to the other tested standards, it has the shortest retention time under reversed-phase liquid chromatography, indicating higher polarity. This suggests it is well-suited for ionization using an electrospray ionization source. Additionally, simultaneous injection of all standards into a charged aerosol detector showed that formaldehyde sodium bisulfite had a similar response to the other standards, with deviations within 20% (Fig. 1). This indicates that it fully elutes from the chromatographic column, supporting its inclusion in the evaluation of camphor-sulfonic acid as a reference standard.

Figure 1. Comparison of the CAD (blue, upper figure) and HRMS (orange, lower figure) response (mean \pm SD) of ten OS standards at an isocratic gradient. The standards were prepared in one batch as an OS-mix. The mix contained: (1) Formaldehyde sodium bisulfite (CH₃SO₄Na), (2) Methyl sulfate sodium salt (CH₃SO₄Na), (3) Sodium ethyl sulfate (C₂H₅SO₄Na), (4) Butyl sulfate sodium salt (C₄H₁₀SO₄Na), (5) Camphor-10-sulfonic acid (C₁₀H₁₆SO₄), (6) 2-hydroxy- α -pinene OS (C₁₀H₁₈SO₅), (7) Sodium octyl sulfate (C₈H₁₇SO₄Na), (8) Sodium n-decyl sulfate (C₁₀H₂₁SO₄Na), (9) Sodium dodecyl sulfate (C₁₂H₂₅SO₄Na), and (10) Tetradecyl sulfate sodium salt (C₁₄H₂₉SO₄Na). Each OS standard had a concentration of 2 ng/ μ L. To test the stability of the instruments, we measured each standard concentration five times. The HPLC flowrate was set to 0.4 mL/min, with 83.5% being directed into the CAD and 16.5% being directed into the MS. The line (solid) is the average response of these compounds with a band (dash) of \pm 20%.

“The application of SPE for enrichment and fractionation of compounds is rarely practiced, but this sample preparation enables unique and novel analytical approaches.” As the other reviewer pointed out, SPE was already used by Gao et al., 2006, JGR-Atmospheres. Furthermore SPE is used to extract HULIS in aerosol samples for quantification and characterization (e.g. Baduel, C., Voisin, D., and Jaffrezo, J. L.: Comparison of analytical methods for Humic Like Substances (HULIS) measurements in atmospheric particles, Atmos. Chem. Phys., 9, 5949–5962, <https://doi.org/10.5194/acp-9-5949-2009>, 2009.). Based on these considerations, I suggest removing “unique” here.

We changed the sentence in line 88-89 (the track-changes version) as follows:

“The application of SPE for enrichment and fractionation of compounds is rarely practiced, but this sample preparation enables ~~unique and~~ novel analytical approaches.”

OS/OM. It is important to clarify in the text, that here the OS mass fraction includes the sulphate group. Meanwhile some of the previous studies listed in Table S7 may have either included or excluded the mass of the sulfate group in the OS fraction. Did the authors check this?

To clarify the fact that we include sulphate group for the OSs mass calculation, we added a sentence in line 630 (the track-changes version, in the figure 5’s caption) as follows:

“We note that the OSs mass fraction includes the sulphate group.”

We also checked the references listed in Table S7, and found that the majority of studies include the sulphate group in the OS/OM calculation. As not all papers provide all the necessary information (e.g. we had to assume an OC to OM conversion factor), therefore, we do not change the table in the SI.

Direct quantification						
location	Particle size	OS fraction in OM (%)	Sulphate group	Instruments	Reference	
Beijing, China	PM _{2.5}	0.31	Include	HPLC/MS	Wang et al., 2018 ²	
Shanghai, China	PM _{2.5}	0.62	Include	UPLC/ToF-MS	Wang et al., 2021 ³	
Beijing (Winter), China	PM _{2.5}	0.7	Include	UHPLC/Orbitrap-MS	Ma et al., 2022 ⁴	
Xi’an (Winter), China	PM _{2.5}	1.4	Include	UHPLC/ToF-MS	Glasius et al., 2022 ⁵	
4 sites, Asia	PM _{2.5}	1.4	Include	UPLC/ ToF-MS	Stone et al., 2012 ⁶	
Beijing, China	PM ₁	2	Include?	FIGAERO ToF-CIMS	Le Breton et al., 2018 ⁷	
19 sites, USA	PM _{2.5}	3.22	Include	UPLC/ToF-MS	Chen et al., 2021 ⁸	
Bohai and Yellow Sea, China	PM _{2.5}	3.47	Include	UHPLC/Orbitrap-MS	Wang et al., 2023 ⁹	
Beijing (Summer), China	PM _{2.5}	4	Include	UHPLC/Orbitrap-MS	Ma et al., 2022 ⁴	
Xi’an (Summer), China	PM _{2.5}	7	Include	UHPLC/ToF-MS	Glasius et al., 2022 ⁵	
Indirect quantification						
Fairbanks, USA	PM _{2.5}	1.29	Exclude?	XRF, IC	Shakya et al., 2013 ¹⁰	
Hongkong, China	PM ₁	5 (Minimum estimation)	Include	HR-ToF-AMS	Huang et al., 2015 ¹¹	
334 sites, USA	PM _{2.5}	7.5	Exclude?	XRF, IC	Shakya et al., 2015 ¹²	
12 sites, USA	PM _{2.5}	8.12	Include	XRF, IC	Tolocka et al., 2012 ¹³	
Riverside, USA	PM _{1.5}	12	Include	AMS, IC	Farmer et al., 2010 ¹⁴	
19 sites, USA	PM _{2.5}	14.51	?	ICP-OES, IC	Chen et al., 2021 ⁸	
Beijing, China	PM _{2.5}	25.05	Include?	XRF, IC	He et al., 2001 ¹⁵	
K-puszta, Hungary	PM _{2.5}	29	Include	XRF, IC	Lukács et al., 2009 ¹⁶	
K-puszta, Hungary	PM _{2.5}	30	Include	XRF, IC	Surratt et al., 2008 ¹⁷	

Reviewer 1's comments regarding response to Reviewer 2

I have now carefully read the responses to point 7 and onwards from the other reviewer.

Overall, I think the authors have addressed the comments well, and this has led to a better understanding of the limitations of the method, as well as suggestions for future work.

7: The authors have thoroughly addressed the issue by performing a new study of isoprene-derived OS and they have added additional text to the manuscript to describe the associated uncertainty. This issue however does not diminish the novelty and overall importance of the study.

8: The authors have correctly studied extraction efficiency and degradation. A recovery below 80% is observed for the largest standard compound. As some of the OS compounds observed in ambient samples are even larger (and thus probably less polar), the authors could mention in the text, that a slightly lower recovery was observed for dodecylsulfate and this may also be relevant for larger OS in general.

9, 10, 11 : The authors have adequately addressed the questions.

12: I agree with the reviewer that it would have been interesting to analyze a well-characterized laboratory-generated OS samples using authentic standards, however the best characterized system is isoprene, which the authors now investigated, but also observed issues with recovery of the OS products. Overall, there is not a standard, reproducible method to generate and verify OS analysis, so the authors have done a good job in addressing the question here. Future work will hopefully lead to an optimized method to study isoprene-derived OS.

Thank you for your thoughtful assessment and for recognizing our efforts in addressing the comments. We thank for your positive feedback on the improvements made to the manuscript, including the clarification of limitations and future research directions.

Regarding point 8, add one sentence in line 142-143 (the track-changes version)

“However, we note that larger, less polar OS compounds may exhibit a reduced extraction efficiency, which should be considered in future studies.”

Once again, we appreciate your time and effort in reviewing our work and providing valuable insights that have helped strengthen the study.